EMBO
Molecular Medicine

# Chronic oxidative stress promotes H2AX protein degradation and enhances chemosensitivity in breast cancer patients

Tina Gruosso[1,2,†], Virginie Mieulet[1,2,†], Melissa Cardon[1,2], Brigitte Bourachot[1,2], Yann Kieffer[1,2], Flavien Devun[3], Thierry Dubois[4], Marie Dutreix[3], Anne Vincent-Salomon[5], Kyle Malcolm Miller[6] & Fatima Mechta-Grigoriou[1,2,*]

## Abstract

Anti-cancer drugs often increase reactive oxygen species (ROS) and cause DNA damage. Here, we highlight a new cross talk between chronic oxidative stress and the histone variant H2AX, a key player in DNA repair. We observe that persistent accumulation of ROS, due to a deficient JunD-/Nrf2-antioxidant response, reduces H2AX protein levels. This effect is mediated by an enhanced interaction of H2AX with the E3 ubiquitin ligase RNF168, which is associated with H2AX poly-ubiquitination and promotes its degradation by the proteasome. ROS-mediated H2AX decrease plays a crucial role in chemosensitivity. Indeed, cycles of chemotherapy that sustainably increase ROS reduce H2AX protein levels in Triple-Negative breast cancer (TNBC) patients. H2AX decrease by such treatment is associated with an impaired NRF2-antioxidant response and is indicative of the therapeutic efficiency and survival of TNBC patients. Thus, our data describe a novel ROS-mediated regulation of H2AX turnover, which provides new insights into genetic instability and treatment efficacy in TNBC patients.

**Keywords** JUND; NRF2; RNF168; Triple-Negative breast cancer; ubiquitination
**Subject Category** Cancer

## Introduction

DNA is permanently damaged by the by-products of metabolism, such as reactive oxygen species (ROS) or by exogenous agents, including radiation and mutagenic chemicals. Leading to significant mutational events such as chromosomal translocations and gene amplifications, DNA damage can cause uncontrolled proliferation and drive malignant transformation (Hoeijmakers, 2001; Bartkova et al, 2005; Gorgoulis et al, 2005; Bonner et al, 2008; Giunta & Jackson, 2011). Cells have therefore developed efficient systems for maintaining their genomic integrity, such as the DNA damage response (DDR). The DDR involves detection of DNA damage, transient cell cycle arrest, damage repair and cell death in case of overwhelming damage (Khanna & Jackson, 2001; Ciccia & Elledge, 2010; Lukas et al, 2011; Polo & Jackson, 2011). Deregulation of the DDR not only compromises genomic integrity in normal cells, favouring tumorigenesis, but also modulates sensitivity to treatment (Bouwman & Jonkers, 2012; Curtin, 2012). Indeed, radiotherapy and chemotherapeutic drugs, such as alkylating agents (cyclophosphamides) or intercalating compounds (anthracyclines), induce DNA damage and subsequently activate the DDR. Thus, DDR is a key guardian of genome integrity, which can also modulate malignant progression by regulating cancer cell sensitivity to cancer therapies involving DNA damage (Bouwman & Jonkers, 2012; Curtin, 2012). The histone variant H2AX is a major component of the DDR, especially functioning in amplifying DNA damage signals (Fernandez-Capetillo et al, 2002, 2004; Bassing et al, 2003; Stucki et al, 2005). Upon DNA damage, phosphatidylinositol 3-kinase (PI-3K) family members (ATM, ATR and DNA-PK) phosphorylate H2AX on Ser139, generating an entity referred to as γ-H2AX (Rogakou et al, 1998). γ-H2AX foci form rapidly at the sites of DNA damage and facilitate the anchoring of signalling and repair proteins, including the MRN complex (NBS1/MRE11/RAD50), 53BP1, MDC1 and RAD51. Consequently, H2AX-deficient cells and mice exhibit genomic instability, suffer from DDR defects and are prone to tumour formation in a p53-null background (Bassing et al, 2002; Celeste et al, 2002, 2003).

Breast cancer (BC) development and invasion are associated with increased ROS production. Accumulation of ROS enhances genomic

1 Stress and Cancer Laboratory, Equipe Labelisée LNCC, Institut Curie, Paris Cedex 05, France
2 Inserm, U830, Paris, France
3 Institut Curie, CNRS UMR3347, INSERM U1021, University Paris-Sud 11, Orsay, France
4 Department of Translational Research, Institut Curie, Paris Cedex 05, France
5 Department of Tumour Biology, Institut Curie, Paris Cedex 05, France
6 Department of Molecular Biosciences, Institute for Cellular and Molecular Biology, University of Texas at Austin, Austin, TX, USA
*Corresponding author. Tel: +33 1 5624 6653; Fax: +33 1 5624 6650; E-mail: fatima.mechta-grigoriou@curie.fr
†These authors contributed equally to this work

instability and significantly modifies the tumour microenvironment (Gerald *et al*, 2004; Martinez-Outschoorn *et al*, 2010; Toullec *et al*, 2010; Balliet *et al*, 2011; Taddei *et al*, 2012). Although oxidative stress promotes tumour growth and spread, it can also improve the response to ROS-producing chemotherapeutic agents (Mateescu *et al*, 2011; Batista *et al*, 2013), an effect that could account for the limited success of antioxidants in clinical trials (Goodman *et al*, 2011). Here, we uncover a new mechanism, by which ROS downregulate H2AX protein levels and sensitize cancer cells to anti-cancer agents. We demonstrate that chronic oxidative stress enhances interaction of the H2AX protein with the E3 ubiquitin ligase RNF168. This consequently favours H2AX protein degradation by the proteasome and results in a steady decrease in H2AX protein level, under chronic accumulation of ROS. H2AX downregulation is detected in several pathophysiological conditions associated with persistent oxidative stress, such as a deficiency of anti-redox sensors, ageing or aggressive BC. Moreover, cycles of chemotherapy, acting in part through increases in ROS concentrations, also significantly reduce H2AX protein levels, in particular in patients for whom the NRF2 antioxidant response is impaired. Interestingly, the decrease in H2AX protein is associated with high rate of tumour cell apoptosis and thus indicative of response to treatment and patient survival, in one of the most aggressive forms of BC, the Triple-Negative (TN) BC subtype. Our work thus identifies a new mechanism by which chronic oxidative stress reduces H2AX protein stability, which provides new insights into genetic instability and is indicative of chemotherapy sensitivity in TNBC patients.

## Results

### Defective JunD-/Nrf2-dependent antioxidant response is associated with reduced H2AX protein levels

To test whether persistent oxidative stress could modulate the DDR, we took advantage of the *junD*-deficient mouse model. JunD regulates genes involved in antioxidant defence and its inactivation

leads to a persistent accumulation of ROS (Gerald *et al*, 2004; Laurent *et al*, 2008; Meixner *et al*, 2010; Toullec *et al*, 2010; Cook *et al*, 2011; Hull *et al*, 2013; Paneni *et al*, 2013). As H2AX is a key component of the DDR, we first investigated the impact of persistent oxidative stress on this protein by Western blot analysis of whole cell extracts from wild-type (*wt*) and *junD$^{-/-}$* fibroblasts (Fig 1A). We observed that the total H2AX protein level was significantly reduced in *junD$^{-/-}$* fibroblasts compared to *wt* (Fig 1A, left). Interestingly, the same downregulation of total H2AX protein levels was also detected in Nrf2-deficient (*Nfe2l2$^{-/-}$*) fibroblasts (Fig 1A, right), another well-known model of chronic oxidative stress (Sporn & Liby, 2012). This observation was consistent with the fact that JunD is required for the full function of the Nrf2 transcription factor and *junD$^{-/-}$* fibroblasts exhibit a significant downregulation of Nrf2-target genes (Bourachot *et al*, In preparation). Similarly, JunD and Nrf2 have been shown previously to interact and mediate a protective response against ROS (Wild *et al*, 1999; Tsuji, 2005; Iwasaki *et al*, 2006; Wang & Jaiswal, 2006; MacKenzie *et al*, 2008; Meixner *et al*, 2010). H2AX decrease was not observed in *catalase*-deficient cells (Ho *et al*, 2004; Ivashchenko *et al*, 2011) (Appendix Fig S1A), indicating that H2AX decrease is tightly associated with defects in JunD/Nrf2-mediated antioxidant defence. We next investigated the levels of other key DDR proteins in *junD$^{-/-}$* and *Nfe2l2$^{-/-}$* fibroblasts by using protein arrays (Appendix Fig S1B and C and Appendix Table S1). Among the proteins tested, H2AX was significantly decreased in both *junD$^{-/-}$* and *Nfe2l2$^{-/-}$* cells, confirming that H2AX protein can be regulated by the deficiency of these transcription factors. Importantly, H2AX downregulation by chronic stress was not restricted to cultured cells but was also detected in organs from *junD$^{-/-}$* mice (Fig 1B) with a marked oxidative stress (Laurent *et al*, 2008; Cook *et al*, 2011; Paneni *et al*, 2013). Interestingly, a significant decrease in the total level of H2AX protein was also observed in tissues of aged *wt* mice, when compared to young mice (Fig 1B). We and others have previously shown that physiological ageing is linked to reduced antioxidant defences and associated with the progressive loss of JunD and Nrf2

---

**Figure 1.  Chronic oxidative stress reduces total H2AX protein levels and prevents normal accumulation of γ-H2AX.**

A   (up) Representative Western blot from whole cell extracts showing H2AX protein levels in *wt* and *junD$^{-/-}$* fibroblasts (left) or in *wt* and *Nfe2l2$^{-/-}$* fibroblasts (right). Two representative clones, with their corresponding controls, are shown per genotype. H2B was used as negative control. (down) Bar plot showing H2AX protein levels as assessed by densitometry analysis of Western blots (as shown above). *n* ≥ 3 independent experiments.

B   (up) Representative Western blot from kidney whole cell extracts showing H2AX protein levels in age-matched *wt* and *junD$^{-/-}$* mice. Mice are 2 months old (young) and 18 months old (old), respectively. (down) Bar plot showing H2AX protein levels as assessed by densitometry analysis of Western blots (as shown above). *N* ≥ 5 mice per age and per genotype.

C   (up) Representative Western blot from kidney whole cell extracts, showing H2AX protein levels in 24 months old *wt* and *junD$^{-/-}$* mice, either untreated (−) or treated with the antioxidant agent, N-acetyl-cysteine (NAC). (down) Bar plot showing H2AX protein levels as assessed by densitometry analysis of Western blots (as shown above). *N* ≥ 4 mice per treatment and per genotype.

D, E   (left) Representative Western blots from whole cell extract showing γ-H2AX, H2AX protein levels and Kap1 phosphorylation (P-Kap1) in *wt* and *junD$^{-/-}$* fibroblasts (D) or in *wt* and *Nfe2l2$^{-/-}$* fibroblasts (E), after H$_2$O$_2$ exposure for the indicated times (hours, h). (right) Bar plots showing γ-H2AX and H2AX protein levels, as well as γ-H2AX/H2AX ratio as assessed by densitometry analysis of Western blots (as shown on the left). *n* = 3 independent experiments.

F, H   (up) Representative Western blots from whole cell extract showing γ-H2AX, H2AX protein levels and P-Kap1 in *wt* and *junD$^{-/-}$* fibroblasts, following 2 Gy of γ-irradiation (F) or after camptothecin (CPT) treatment (H) for the indicated times (hours, h). (down) Bar plots showing γ-H2AX and H2AX protein levels, as well as γ-H2AX/H2AX ratio as assessed by densitometry analysis of Western blots (as shown upper). *n* = 3 independent experiments.

G, I   (up) Representative γ-H2AX immunofluorescence (green) in *wt* and *junD$^{-/-}$* fibroblasts before (−) or after γ-irradiation (2 Gy, 45 min) (G) or after camptothecin (CPT) treatment for 1 h (I). Blue signal corresponds to DAPI staining. (down) box-plots of large γ-H2AX foci per nuclei (diameter > 0.8 μm). At least 50 nuclei per genotype have been used for quantification.

Data information: For all panels, data are shown as means ± SEM. *P*-values are based on Student's *t*-test (A–C, G and I). *P*-values show the difference between *wt* and *junD$^{-/-}$* or between *wt* and *Nfe2l2$^{-/-}$* fibroblasts across time points and are based on paired *t*-test (D–F and H). Actin is used as internal control for protein loading (A–F and H). NS stands for not significant and * stands for *P*-value = 0.01. Scale bars = 10 μm. a.u. stands for arbitrary unit.

Source data are available online for this figure.

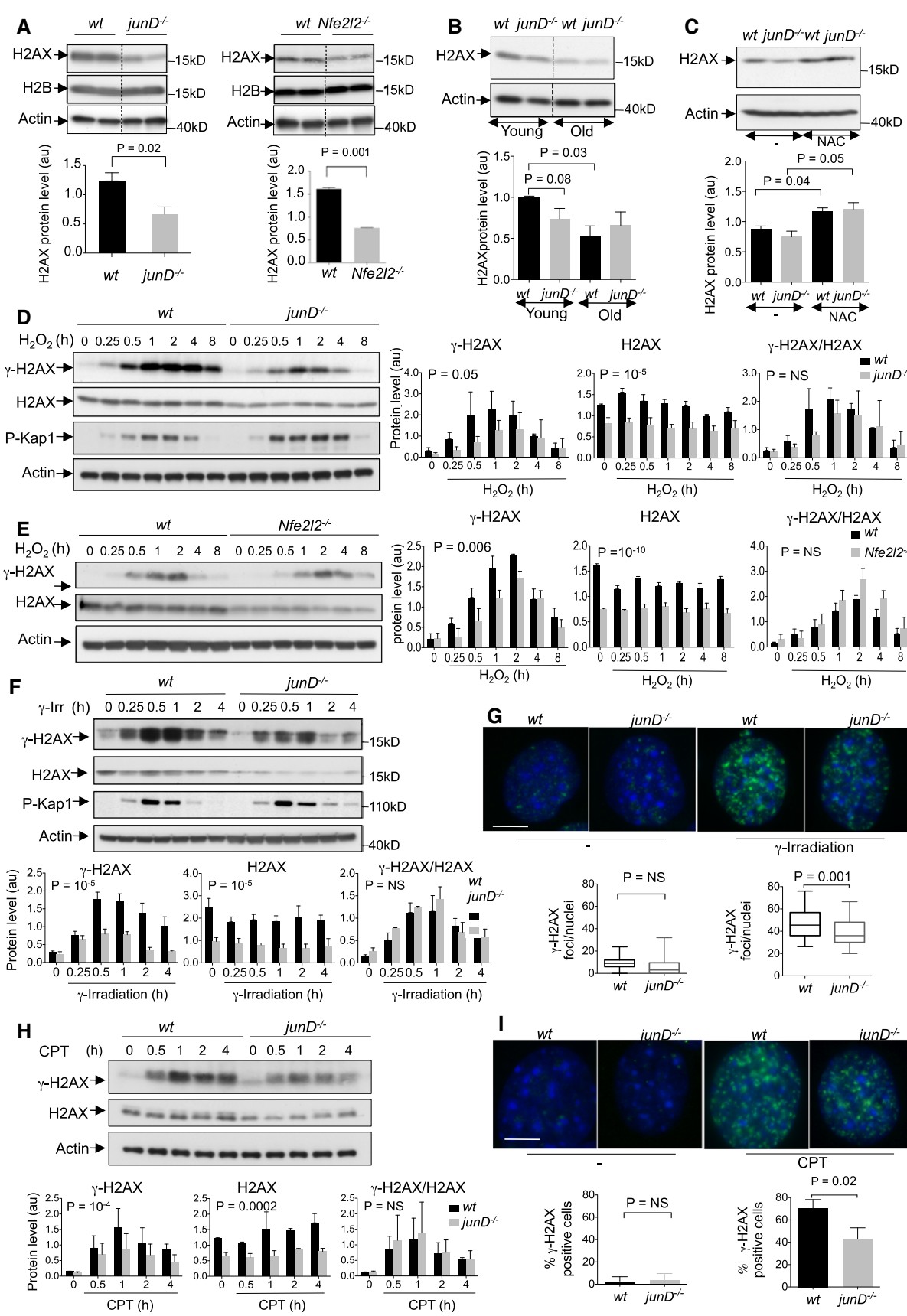

**Figure 1.**

transcription factors (Medicherla *et al*, 2001; Bokov *et al*, 2004; Suh *et al*, 2004; Shih & Yen, 2007; Laurent *et al*, 2008; Sykiotis & Bohmann, 2010; Toullec *et al*, 2010; Cook *et al*, 2011; Miller *et al*, 2012; Sena & Chandel, 2012; Paneni *et al*, 2013; Rahman *et al*, 2013; Tome *et al*, 2014). Thus, our observations emphasized the potential role of these transcription factors in the regulation of H2AX protein. As $junD^{-/-}$ mice age prematurely, downregulation of H2AX protein was observed at a young age in $junD^{-/-}$ mice while it occurred later in *wt* mice (Fig 1B). Finally, to assess that the reduction in the total level of H2AX protein observed in young $junD^{-/-}$ mice and old *wt* mice was a consequence of ROS accumulation, we treated these mice with an antioxidant drug, N-Acetylcysteine (NAC) (Fig 1C). H2AX protein levels were restored in NAC-treated $junD^{-/-}$ mice to similar levels as in their *wt* counterparts (Fig 1C). Thus, long-term antioxidant treatment prevented H2AX downregulation in a genetic model ($junD^{-/-}$) and in a physiological process (ageing) of chronic oxidative stress, supporting that ROS are involved in regulating the total level of H2AX protein *in vivo*.

## Downregulation of H2AX prevents normal accumulation of γ-H2AX following acute stress

To decipher the link between H2AX and oxidative stress, we next analysed H2AX protein levels following genotoxic stresses in cells suffering from chronic oxidative stress (Fig 1D–I). As expected, exposure of *wt* cells to $H_2O_2$ strongly stimulated H2AX phosphorylation (Fig 1D). However, H2AX was less efficiently phosphorylated following $H_2O_2$ exposure in *junD*-deficient cells than in *wt* cells (Fig 1D). This regulation was specific to H2AX, as the phosphorylation of Kap1, another well-known ATM target (Goodarzi *et al*, 2008; Noon *et al*, 2010), was not decreased by *junD* depletion (Fig 1D). Moreover, γ-H2AX/H2AX ratios were equivalent in $junD^{-/-}$ and *wt* fibroblasts (Fig 1D, right panels). These observations suggest that ATM, ATR or DNA-PK were similarly active in *wt* and $junD^{-/-}$ cells and that the reduced levels of phosphorylated H2AX in $junD^{-/-}$ cells were due to the downregulation of its total protein. Same observations were made in *Nrf2*-deficient cells, with a significant decrease in H2AX phosphorylation after acute stress, resulting

from reduced total H2AX protein levels (Fig 1E). This argues that H2AX follows similar regulation in $Nfe2l2^{-/-}$ fibroblasts than in $junD^{-/-}$ cells, consistent with the fact that Nrf2 activity is impaired in *junD*-deficient cells (Meixner *et al*, 2010; Bourachot *et al*, In preparation). Finally, we also observed reduced H2AX phosphorylation rate in $junD^{-/-}$ cells following genotoxic stresses (Fig 1F–I). Indeed, upon γ-irradiation (Fig 1F and G) or camptothecin application (Fig 1H and I), the decrease in the total level of H2AX protein prevented the normal accumulation of its phosphorylated form γ-H2AX (Fig 1F and H) and the formation of γ-H2AX foci (Fig 1G and I) in $junD^{-/-}$ cells. Thus, downregulation of H2AX protein in cells suffering from chronic oxidative stress prevents the normal accumulation of γ-H2AX following acute stress that could result in an increased sensitivity to DNA-damaging agents.

## Reduced H2AX protein level is associated with increased DNA damage

Upon DNA damage, normal cells undergo cell cycle arrest until the repair of DNA lesions is complete, or cell death in case of overwhelming damage (Khanna & Jackson, 2001; Ciccia & Elledge, 2010; Lukas *et al*, 2011; Polo & Jackson, 2011). We thus checked whether the reduced total level of H2AX protein observed in $junD^{-/-}$ cells could be associated with an abnormal cell cycle and/or survival under normal or DNA-damaging conditions (Fig 2). Compared to *wt* cells, *junD*-deficient fibroblasts exhibited a longer doubling time (Fig 2A). This resulted from an accumulation of cells in the G1-phase of the cell cycle and a concomitant delayed entry into the S-phase (Fig 2B and C). Evaluation of DNA damage by COMET assay showed increased spontaneous DNA damage in $junD^{-/-}$ fibroblasts compared to *wt* cells (Fig 2D–H). At basal state, while only 3% of *wt* fibroblasts exhibited tail moments higher than 5, this proportion reached 16% for $junD^{-/-}$ cells ($P = 0.03$ by Fisher's exact test) (Fig 2D and E). After γ-irradiation (t0), tail moments were highly increased both in *wt* and $junD^{-/-}$ fibroblasts (Fig 2D, F and G), but the distribution of cells with high DNA damage remained significantly higher in $junD^{-/-}$ cells at all time points (Fig 2F and G). Consistently, using clonogenic assays,

**Figure 2. *junD*-deficient fibroblasts exhibit cell cycle delay, decreased cell survival and increased DNA damage.**

A   Bar plot showing the doubling time of *wt* and $junD^{-/-}$ fibroblasts.
B   Representative cell cycle distribution of asynchronous *wt* and $junD^{-/-}$ fibroblasts.
C   Bar plots showing the percentage of *wt* and $junD^{-/-}$ fibroblasts in G1, S and G2 phases of the cell cycle, following serum stimulation for the indicated times (hours, h), after 48-h serum starvation.
D   Representative views of alkaline comet assays from *wt* and $junD^{-/-}$ fibroblasts either untreated or after 10 Gy γ-irradiation for the indicated times (minutes, min).
E   Alkaline comet assays from *wt* and $junD^{-/-}$ fibroblasts at basal state. (left) Box-plots showing tail moments at basal state in *wt* and $junD^{-/-}$ fibroblasts. The black lines of the box-plots represent the median values and the whiskers represent the 10th and the 90th percentiles. (right) Distribution of *wt* and $junD^{-/-}$ fibroblasts expressed as percentage of cells (%) according to the categorized tail moments. At basal state, 16% of $junD^{-/-}$ cells exhibit tail moment higher than 5, while it is the case for only 3% of *wt* cells.
F   Alkaline comet assays from *wt* and $junD^{-/-}$ fibroblasts after 10 Gy γ-irradiation. Box-plots showing tail moments at basal state in *wt* and $junD^{-/-}$ fibroblasts, following 10 Gy γ-irradiation for the indicated times (minutes, min). The black lines of the box-plots represent the median values and the whiskers represent the 10th and the 90th percentiles.
G   Distribution of *wt* and $junD^{-/-}$ fibroblasts expressed as percentage of cells (%) according to the categorized tail moments, following 10 Gy γ-irradiation for the indicated times (minutes, min).
H   Surviving fraction of *wt* and $junD^{-/-}$ fibroblasts 7 days after 2 Gy or 5 Gy γ-irradiation, as assessed by clonogenic assay. Surviving fraction is expressed as percentage (%) of number of colonies after irradiation normalized with respect to untreated conditions.
I   Surviving fraction of *wt* and $junD^{-/-}$ fibroblasts either transfected with an empty vector (+ Ctl) or H2AX-expressing (+ H2AX) vector. Surviving fraction is expressed as percentage (%) of alive cells, 72 h post-transfection.

Data information: For all panels, data are shown as means ± SEM. $n = 3$ independent experiments (A–C, H and I) and at least 100 nuclei per genotype per condition have been analysed for comet assays (E–G). *P*-values are based on Student's *t*-test (A, C, E left, F, H and I) and on Fisher's exact test (E right, and G).

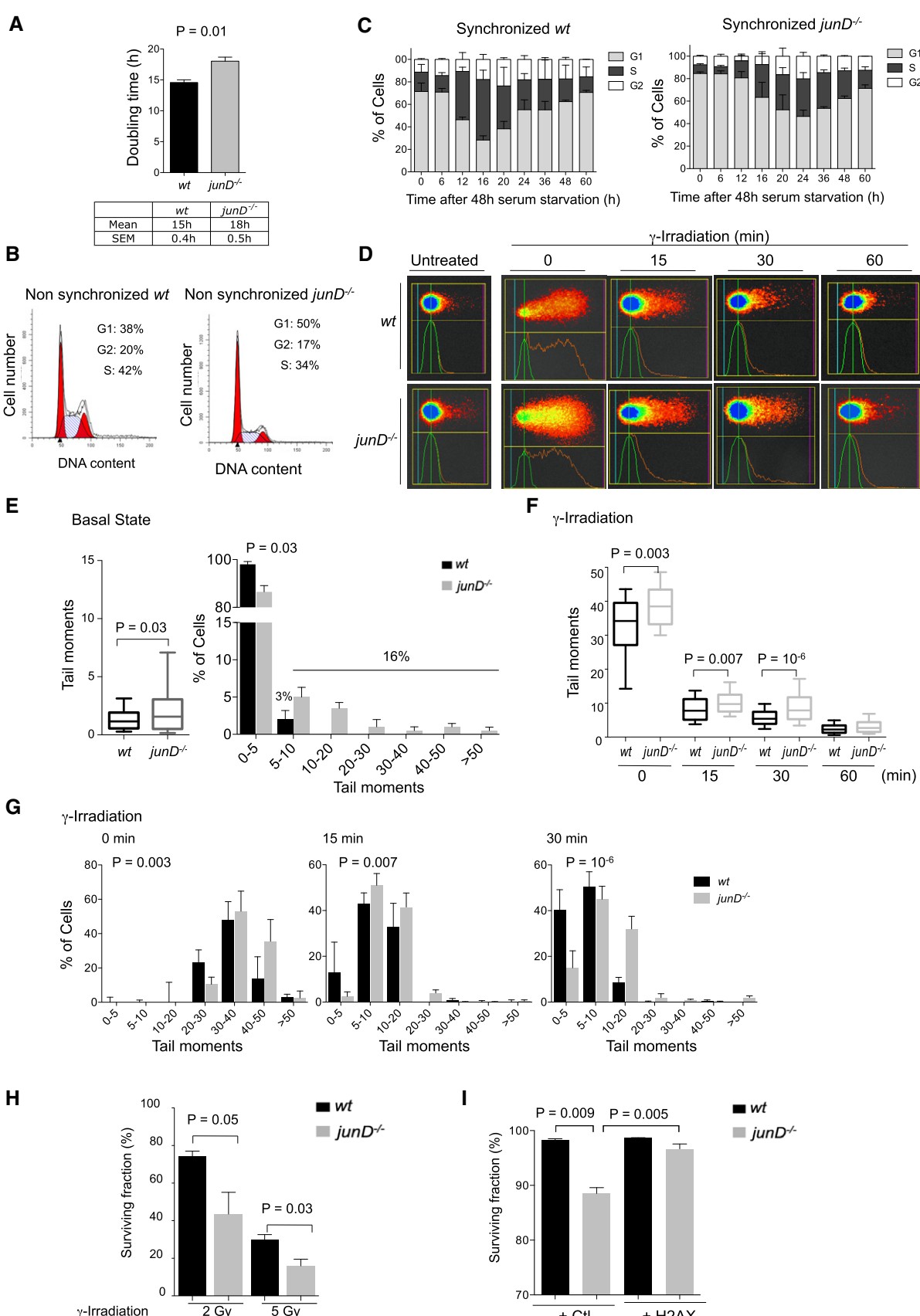

Figure 2.

we observed that $junD^{-/-}$ fibroblasts exhibited a reduced survival rate after γ-irradiation compared to *wt* cells (Fig 2H), as it was demonstrated in previous studies for H2AX haploinsufficiency, H2AX knockout (KO) or after expression of a miRNA targeting H2AX (Bassing *et al*, 2002, 2003; Celeste *et al*, 2002, 2003; Meador *et al*, 2008; Revet *et al*, 2011; Wang *et al*, 2011). Finally, H2AX re-expression in $junD^{-/-}$ fibroblasts was sufficient to rescue their survival rate (Fig 2I), showing that the reduced survival rate observed for $junD^{-/-}$ cells is at least in part due to H2AX protein decrease. In conclusion, *junD*-deficient fibroblasts exhibit reduced survival rate and enhanced sensitivity to γ-irradiation, due to increased DNA damage at basal state and following irradiation, possibly due to low H2AX levels in $junD^{-/-}$ cells.

**Chronic oxidative stress promotes H2AX protein degradation by the proteasome**

We next sought to gain insight into the mechanism by which persistent oxidative stress reduces H2AX protein level. As mentioned above, *junD*-deficient fibroblasts exhibited an increased doubling time due to cell cycle retardation (Fig 2A–C); therefore, the reduced H2AX protein level in *junD*-deficient fibroblasts could not be due to a faster proliferation rate compared to *wt* cells. Moreover, we observed similar levels of *H2afx* mRNA in $junD^{-/-}$ and *wt* fibroblasts (Appendix Fig S2A), as well as the same expression of several miRNA known to target H2AX, such as miR-24 and miR-138

(Lal *et al*, 2009; Wang *et al*, 2011), (Appendix Fig S2B) or predicted to target H2AX (Appendix Fig S2C). These observations suggest that the regulation of H2AX neither occurs at transcriptional nor at post-transcriptional level. We thus tested whether chronic oxidative stress could affect H2AX protein synthesis or stability by using specific inhibitors (cycloheximide and MG132, respectively) and c-Jun protein as positive control for ubiquitination-dependent degradation (Musti *et al*, 1996). We were unable to detect any effect of these inhibitors on H2AX protein levels, by analysing whole cell extracts enriched in chromatin-associated H2AX, while clear effects were detected on the Jun protein, used as a positive control (Appendix Fig S2D and E). We hypothesized that the H2AX protein might be too stable, when associated with chromatin, to detect any effects of these treatments on H2AX half-life, at short time points (up to 8 h) compatible with cell survival. We thus next performed experiments using protein extracts from chromatin-free fraction, as previously described (Ikura *et al*, 2007; Rios-Doria *et al*, 2009). We first observed that the decrease in the total level of H2AX protein was occurring in the free fraction protein extracts of $junD^{-/-}$ or $Nfe2l2^{-/-}$ cells, as it was in whole cell extracts (Fig 3A; Appendix Fig S2F), indicating that the reduced level of H2AX in these cells could not only be due to an altered incorporation into the chromatin. While inhibition of protein synthesis had no obvious effect on H2AX (Appendix Fig S2G), inhibition of the proteasome significantly increased H2AX protein levels both in *wt* and $junD^{-/-}$ cells (Fig 3B), but interestingly, to a higher extent in $junD^{-/-}$ cells

**Figure 3.  Persistent oxidative stress promotes proteasome-dependent degradation of H2AX mediated by RNF168 and ubiquitination of K119.**

A  (up) Representative Western blot from chromatin-free fraction or whole cell extracts, as indicated, showing H2AX protein levels in *wt* and $junD^{-/-}$ fibroblasts. Different amounts of loaded proteins and exposures (low or high exp) are presented, H2AX being easier to detect when associated with chromatin. (down) Bar plots showing H2AX protein levels as assessed by densitometry analysis of Western blots (as shown upper).

B  (up) Representative Western blot from chromatin-free fraction extracts showing H2AX protein levels in *wt* and $junD^{-/-}$ fibroblasts either untreated (−) or after 8 h treatment (+) with proteasome inhibitor MG132. Jun is used as positive control for treatment efficiency. (down) Bar plots showing H2AX protein levels as assessed by densitometry analysis of Western blots (as shown upper).

C  Bar plot showing stabilization of H2AX protein in *wt* and $junD^{-/-}$ fibroblasts, as assessed by densitometry analysis of Western blots (as shown in B). Data are expressed as percentage (%) of H2AX protein levels after treatment with MG132 with respect to untreated conditions.

D  Representative Western blots from whole cell extracts showing HA-tagged ubiquitin (left) or Flag-tagged H2AX (right) protein levels. *wt* and $junD^{-/-}$ fibroblasts were co-transfected with vectors encoding HA-tagged ubiquitin and Flag-tagged wild-type H2AX (WT), or K13-, K15-, K119-, or 9K-H2AX mutants. Flag-tagged H2AX proteins were then immunoprecipitated with Flag-specific antibody and incubated either with HA-specific antibody (left) or Flag-specific antibody (right). Two different exposures (low or high exp) are presented. * stands for the immunoglobulins.

E  (left) Representative Western blots showing H2AX-WT or H2AX-K119 protein levels (revealed using Flag-specific antibody) from whole cell extracts in *wt* and $junD^{-/-}$ fibroblasts. Cells were first transfected with Flag-tagged wild-type H2AX (H2AX-WT) or K119 H2AX mutant (H2AX-K119) and next treated with cycloheximide (CHX) for the indicated times (hours, h). (right) Bar graph showing H2AX-WT and H2AX-K119 protein half-life in *wt* and $junD^{-/-}$ fibroblasts, as indicated. Protein half-life has been calculated from the degradation curve of H2AX protein (based on densitometry analysis of Western blots, as shown on the left) by extrapolating its linear part.

F  (left) Representative Western blots showing H2AX-WT protein levels (revealed using Flag-specific antibody) from whole cell extracts in $junD^{-/-}$ fibroblasts co-transfected with Flag-tagged H2AX-WT construct and either non-targeting siRNA (siCtrl) or siRNA targeting RNF168 (siRNF168), RNF8 (siRN8) or BMI1 (siBMI1). Efficiency of each siRNA has been verified and reached 60% of inhibition, in average, as shown for RNF168 (up). Protein extracts were processed following cycloheximide (CHX) treatment for the indicated times (hours, h). (right) Bar graph showing the percentage (%) of decrease in H2AX-WT protein levels after 4 h treatment with CHX ($t = 4$) with respect to untreated conditions ($t = 0$) in $junD^{-/-}$ fibroblasts transfected with specific siRNA, as indicated.

G  (left) Representative Western blots showing H2AX-WT or H2AX-K119 protein levels from whole cell extracts in $junD^{-/-}$ fibroblasts transfected either with Flag-tagged wild-type H2AX (H2AX-WT) or K119 H2AX mutant (H2AX-K119) and co-transfected with non-targeting siRNA (siCtrl) or siRNA targeting RNF168 (siRNF168). Cells were treated with cycloheximide (CHX) for the indicated times (hours, h) before protein extraction. (right) Bar graph showing H2AX-WT and H2AX-K119 protein half-life in $junD^{-/-}$ fibroblasts transfected with non-targeting siRNA (siCtrl) or siRNA targeting RNF168 (siRNF168). Protein half-life has been calculated from the degradation curve of H2AX protein (based on densitometry analysis of Western blots, as shown on the left) by extrapolating its linear part.

H  Representative Western blot showing RNF168, Flag-tagged H2AX (H2AX-FLAG) and endogenous H2AX (H2AX) protein levels from whole cell extracts in *wt* and $junD^{-/-}$ fibroblasts. Cells were first co-transfected with vectors encoding HA-tagged ubiquitin and Flag-tagged wild-type H2AX (WT), or K13-, K15-, K119-, or 9K-H2AX mutants. Flag-tagged H2AX proteins were next immunoprecipitated with Flag-specific antibody and incubated either with RNF168-specific antibody (up) or H2AX-specific antibody (down).

Data information: For all panels in this figure, data are means ± SEM. ($n = 3$ independent experiments). *P*-values are based on Student's *t*-test. NS stands for not significant. Actin is used as internal control for protein loading (A and B).

Source data are available online for this figure.

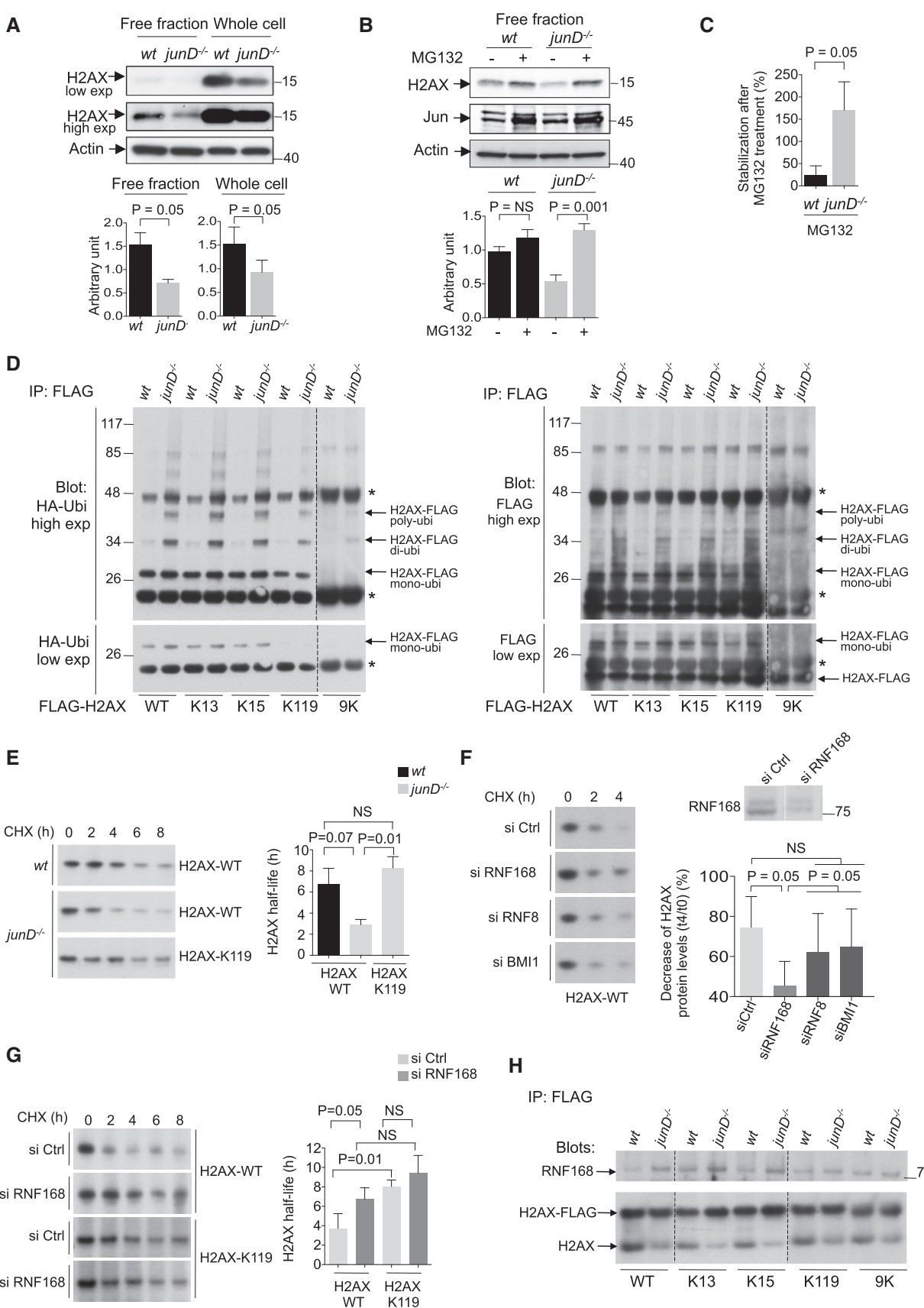

**Figure 3.**

(Fig 3C). These data not only confirm that H2AX is degraded in the nucleoplasmic fraction, as previously shown (Rios-Doria et al, 2009), but also show that H2AX protein is more degraded in $junD^{-/-}$ cells than in wt cells.

## RNF168 mediates H2AX poly-ubiquitination and degradation under chronic oxidative stress

We next aimed at identifying the molecular process leading to proteasome-dependent degradation of H2AX under persistent oxidative stress. Ubiquitin conjugation on lysine residues of the target protein is typically associated with proteasome-dependent degradation.

We thus analysed the ubiquitination state of H2AX in wt and junD-deficient fibroblasts (Fig 3D–G; Appendix Fig S2H) by taking advantage of epitope-tagged forms of H2AX, which have been previously shown to behave and be regulated in the same manner as H2AX endogenous protein (Rios-Doria et al, 2009; Chen et al, 2013; Leung et al, 2014). H2AX contains nine surface-exposed lysine residues including K13, K15 and K119 that have previously been identified as ubiquitination sites that are critical for DNA damage signalling (Chen et al, 2013; Leung et al, 2014; Kocylowski et al, 2015). We thus compared ubiquitination patterns of the wild-type form of H2AX (H2AX-WT) with H2AX mutated versions that contain individual K to A mutations (i.e. K13, K15, K119) or lacking modifiable lysines (i.e. 9K) in both wt and junD-deficient fibroblasts (Fig 3D, see also controls with empty vector and FLAG-H2AX-expressing plasmid in Appendix Fig S2H). Interestingly, we observed an increased level of mono-, di- and poly-ubiquitinated H2AX in $junD^{-/-}$ compared to wt cells, when expressing the H2AX-WT synthetic protein. This ubiquitination pattern was reduced when expressing either H2AX-K119 or H2AX-9K mutated forms, while K13 or K15 mutants had no impact (Fig 3D), suggesting that K119 is specifically required for H2AX poly-ubiquitination in $junD^{-/-}$ cells. Although the role of histone ubiquitination has been well established in DNA damage signalling, it was unclear whether it could also regulate H2AX stability. We thus investigated whether poly-ubiquitination of H2AX on K119 might affect H2AX protein stability (Fig 3E). We treated cells with cycloheximide to inhibit de novo protein synthesis and measure H2AX half-life by monitoring protein disappearance (See Materials and Methods). We first observed that H2AX-WT protein exhibited a shorter half-life in $junD^{-/-}$ fibroblasts compared to wt cells (Fig 3E). Interestingly, mutation of the K119 residue was sufficient to rescue H2AX protein stability in $junD^{-/-}$ cells (Fig 3E). Indeed, H2AX-K119 half-life in $junD^{-/-}$ cells was equivalent to H2AX-WT protein in wt cells, indicating that K119 residue was crucial for regulating H2AX stability in $junD^{-/-}$ cells. These results were in agreement with previous studies showing that ubiquitination of K119 residue in H2AX tail could mediate H2AX degradation (Rios-Doria et al, 2009). We then looked for E3 ubiquitin ligases that might be implicated in this process. In particular, the E3 ubiquitin ligases RNF8, RNF168 and RING1B/BMI1 that are known to ubiquitinate H2AX on K13/15 and K119, respectively, although these enzymes have until now been strictly involved in the activation of the DDR (Kolas et al, 2007) (Doil et al, 2009; Pinato et al, 2009; Stewart et al, 2009; Ismail et al, 2010; Ginjala et al, 2011; Gatti et al, 2012; Mattiroli et al, 2012; Bohgaki et al, 2013; Chen et al, 2013; Fradet-Turcotte et al, 2013; Jackson &

Durocher, 2013; Leung et al, 2014; Kocylowski et al, 2015). We thus evaluated whether the silencing of any of these enzymes might affect H2AX-WT protein stability. Inactivation of each gene reached at least 60% of efficiency in average, as shown for RNF168 (Fig 3F and Appendix Fig S2I). Interestingly, RNF168 was the only enzyme, whose inactivation significantly stabilized H2AX-WT protein in $junD^{-/-}$ cells (Fig 3F). Indeed, RNF168 silencing in $junD^{-/-}$ cells mimicked K119 mutation regarding H2AX protein half-life (Fig 3G), indicating that the E3 ubiquitin ligase, RNF168, was a key regulator for H2AX stability in these cells. In agreement with these observations, we detected an enhanced interaction of RNF168 with H2AX-WT protein in $junD^{-/-}$ fibroblasts, compared to wt cells (Fig 3H). This enhanced interaction was conserved in cells expressing either H2AX-K13 or H2AX-K15 mutant proteins, indicating that these residues were not critical for H2AX interaction with RNF168 in $junD^{-/-}$ fibroblasts. In contrast, RNF168 interaction with H2AX-K119 or H2AX-9K mutants was significantly reduced in $junD^{-/-}$ cells, reaching the same level of interaction as the one observed in wt cells (Fig 3H). Thus, enhanced interaction between RNF168 and H2AX in $junD^{-/-}$ cells was mediated through the K119 residue, an ubiquitination site that controls H2AX half-life. Taken as a whole, these data show that chronic oxidative stress due to junD loss increases RNF168-mediated ubiquitination of H2AX on K119, ultimately leading to H2AX degradation by the proteasome. These observations thus suggest that the E3 ubiquitin ligase RNF168 not only participates in DDR signalling through ubiquitination of K13/K15 as shown previously (Mattiroli et al, 2012), but is also involved in H2AX stability under persistent oxidative stress.

## Low H2AX protein levels in Triple-Negative breast cancers (TNBC) are associated with oxidative stress signatures

We next examined the physiological relevance of H2AX downregulation by chronic oxidative stress. As junD- or Nfe2l2-deficient fibroblasts mimic features of carcinoma-associated fibroblasts (CAFs) due to ROS increase (Toullec et al, 2010; Artaud-Macari et al, 2013), we first tested whether H2AX downregulation could be observed in CAFs from invasive breast carcinomas. H2AX protein levels were analysed by immunohistochemistry (IHC) in three classes of human BC with distinct clinical outcomes and myofibroblast content: Luminal-A (LumA), ERBB2-amplified (HER2) and Triple-Negative (TN) tumours (Fig 4; Table 1 for cohort description). H2AX protein was clearly detected in fibroblasts and epithelial cells of LumA tumours (Fig 4A and B). In contrast, H2AX levels dropped in both compartments (Fig 4A and B), in a correlative way (Fig 4C), in HER2 and TN carcinomas, which have been characterized by chronic oxidative stress (Finak et al, 2008; Martinez-Outschoorn et al, 2010; Toullec et al, 2010; Balliet et al, 2011; Parri & Chiarugi, 2013). Histological staining of actin, alpha2, smooth muscle (SMA), a marker of activated fibroblast upregulated by persistent oxidative stress (Martinez-Outschoorn et al, 2010; Toullec et al, 2010; Balliet et al, 2011; Artaud-Macari et al, 2013) accumulated in CAF from HER2 and TN tumours compared to LumA tumours (Fig 4D and E) and was negatively correlated with H2AX protein levels (Fig 4F). Together with reduced H2AX protein levels, aggressive BC subtypes have higher proliferation rate, as evaluated by the mitotic index (Fig 4G). There was a faint inverse correlation between both parameters when considering all BC subtypes

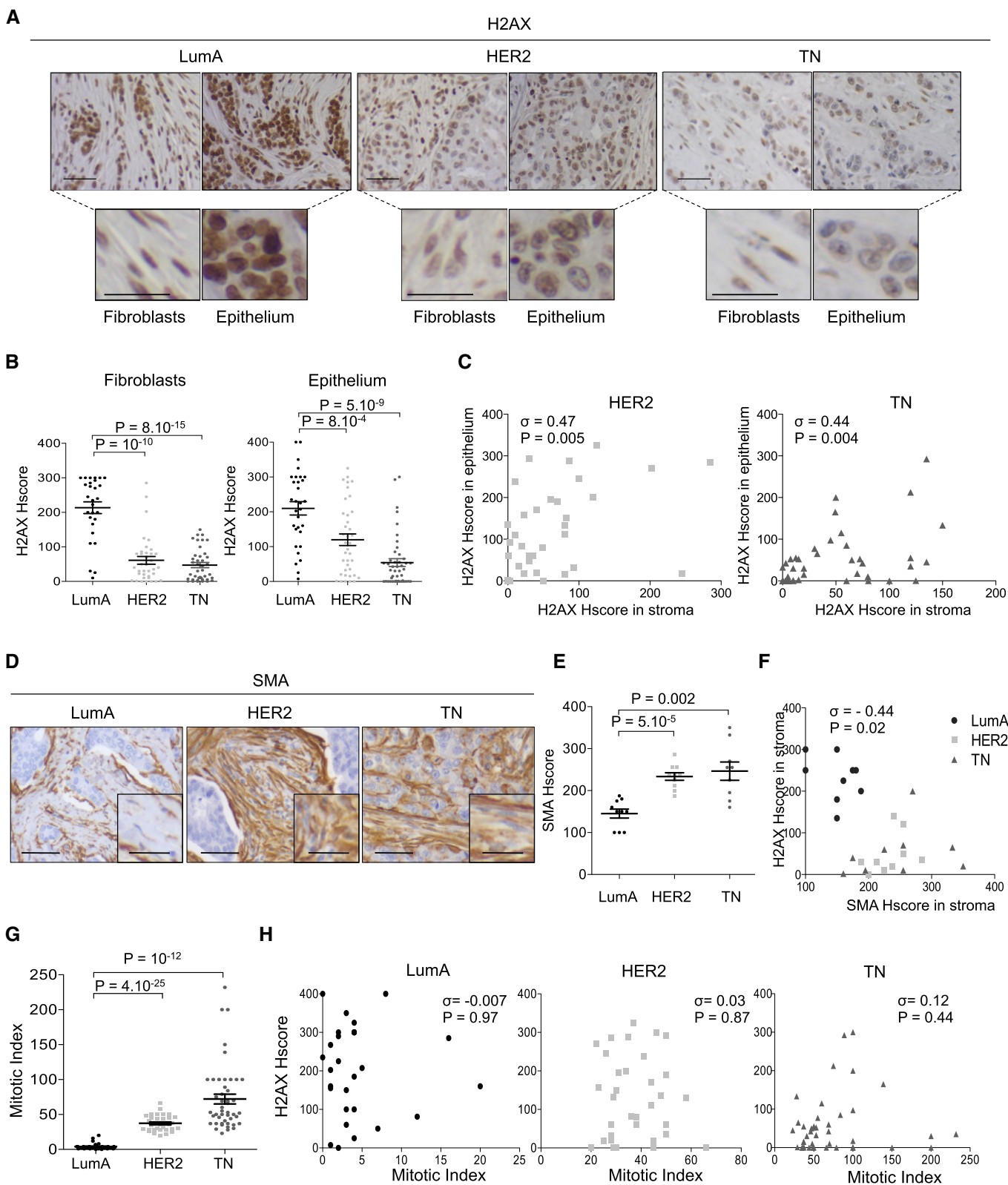

Figure 4.

collectively ($\rho = -0.25$, $P = 0.01$ by Spearman's test). However, while H2AX protein levels were highly variable within each BC subtype (as shown Fig 4B, right), H2AX protein levels did not correlate with the mitotic index when considering each BC subtype separately (Fig 4H), suggesting a mechanism other than proliferation within each BC subtype. Moreover, H2AX protein levels were significantly

**Figure 4.  H2AX staining is reduced both in stroma and epithelium of aggressive BC.**

A   (up) Representative views of H2AX immunostaining from 117 BC patients of Luminal-A (LumA, N = 33), HER2 (N = 37) and Triple-Negative (TN, N = 47) subtypes.
    Scale bars = 100 μm. (down) representative views at high magnification of H2AX immunostaining focused on the Fibroblasts (left) or Epithelium (right) cells. Scale
    bars = 25 μm.
B   Scatter plots of H2AX histological score (Hscore = staining intensity (0–4) × % of positive cells quantified from H2AX immunostaining, as shown in A) in Fibroblasts
    (left) or Epithelial (right) cells from LumA, HER2 and TN breast tumours.
C   Correlation plots between H2AX Hscores evaluated in fibroblasts and in epithelial cells (as shown in B), in HER2 (left) and TN (right) breast tumours.
D   Representative views of actin, alpha2, smooth muscle (SMA) immunostaining from BC patients (N = 29) of Luminal-A (LumA, N = 10), HER2 (N = 10) and TN (N = 9)
    subtypes. Scale bars = 100 μm. Zoom views are focused on the fibroblast compartment. Scale bars = 25 μm.
E   Scatter plots of SMA histological score (Hscore) evaluated from SMA immunostaining (as shown in D).
F   Correlation plot between SMA and H2AX Hscores in stromal compartment.
G   Scatter plot of the mitotic index, evaluated by pathologists, in LumA, HER2 and TNBC patients, as indicated.
H   Absence of correlation between mitotic index and epithelial H2AX Hscore, in LumA HER2 and TNBC.

Data information: Data are shown as mean ± SEM (B, E and G). P-values are based on Welch's t-test (B) or Student's t-test (E and G). Correlation coefficients σ and
P-value are based on Spearman's rank correlation test (C, F and H).

correlated between the epithelial and stromal compartments in HER2 and TN patients (as shown above, Fig 4C), while stromal cells proliferate at a much lower rate than tumour cells. Thus, proliferation could not be the sole mechanism regulating H2AX protein level. We thus sought to define the transcriptomic signatures enabling us to distinguish the tumours with high- versus low-H2AX protein levels (Welch's t-test) in aggressive BC subtype. Interestingly, the "stress response" signatures were significantly enriched in tumours with low-H2AX protein levels both in TN and HER2 subtypes (TN tumours: Fold enrichment = 2.1; $P = 3 \times 10^{-7}$ by Welch's t-test; Appendix Table S2, for gene list. HER2 tumours: Fold enrichment = 2.4; $P = 3 \times 10^{-11}$ by Welch's t-test; Appendix Table S3, for gene list). In particular, genes encoding NADPH oxidases were highly upregulated in aggressive BC with low-H2AX protein levels (TN: Fold enrichment = 35.7; $P = 1 \times 10^{-4}$; HER2: Fold enrichment = 24.2; $P = 0.006$ by Hypergeometric test). As no signature related to cell division was found, this suggests that proliferation rate does not contribute to the regulation of H2AX within HER2 or TNBC subtype and supports a ROS-mediated downregulation of H2AX in aggressive human BC.

**H2AX downregulation by chemotherapy is indicative of response to treatment and survival of TN patients**

We next considered whether chemotherapy, whose successive cycles increase ROS in a chronic manner, could also modulate H2AX protein levels and subsequently the γ-H2AX DNA damage signal. By gene ontology analyses, we confirmed that the chemotherapy administered to this cohort of TN patients (see Table 2 for cohort description) induced a specific response to oxidative stress (Fold enrichment = 2.1; $P = 0.001$ by Hypergeometric test; Appendix Table S4, for gene list). Interestingly, there was also a significant enrichment in NRF2-dependent transcriptomic signature (Malhotra et al, 2010; Singh et al, 2013; Hayes & Dinkova-Kostova, 2014) in these TN patients after chemotherapy (Fig 5A), indicating that chemotherapy induced acute oxidative stress, which promoted an NRF2-mediated antioxidant response. As TN patients display variable response to neo-adjuvant chemotherapy, we wondered whether variations in oxidative stress response and H2AX protein levels could play a role in this process. We thus analysed the effect of chemotherapy on H2AX protein levels in TN patients. H2AX protein levels and γ-H2AX were significantly reduced in residual tumours of TN patients after chemotherapy (Fig 5B). H2AX

downregulation accounted for the decrease in γ-H2AX, as the γ-H2AX/H2AX ratio remained equivalent (Fig 5B), confirming, in human tumours, the observations we made in genetic models of chronic oxidative stress (Fig 1). Several previous studies have nicely demonstrated that reduced amount of H2AX increases DNA damage and sensitivity to anti-cancer therapies (Bassing et al, 2003; Celeste et al, 2003; Meador et al, 2008; Revet et al, 2011). We confirmed this observation on BC cell lines using cisplatin, a DNA cross-linking agent routinely used for cancer treatment (Fig 5C). Indeed, we found a positive correlation between H2AX protein levels and sensitivity of various BRCA1/2 wild-type (Elstrodt et al, 2006) BC cell lines to cisplatin (Fig 5C). Moreover, we observed that partial inactivation of H2AX by siRNA was sufficient to sensitize TNBC cells to cisplatin-induced apoptosis (Fig 5D and E). Importantly, the extent of H2AX downregulation in TN patients following chemotherapy was significantly correlated with their pathological response to therapy (Fig 5F). In contrast, H2AX variation in TNBC patients after chemotherapy was not associated with any other clinical features evaluated at diagnosis, including mitotic status, histological grade, axillary or distant metastases (NM status). Indeed, we found no significant association between H2AX downregulation and any of these clinical parameters, as evaluated by Fisher's exact test. Interestingly, the extent of H2AX decrease after chemotherapy was indicative of overall patient survival and progression-free survival, as shown by a Kaplan–Meier survival test (Fig 5G) and confirmed using univariate Cox proportional hazard model (HR = 1.23; CI 95% Inf = 1.02; CI 95% Sup = 1.5; $P = 0.03$ by Cox proportional hazard regression). Consistent with this observation, tumours with major-H2AX decrease showed higher number of apoptotic tumour cells, compared to tumours with minor-H2AX decrease (Fig 5H). As NRF2-dependent signature was significantly enriched following chemotherapy, we next tested whether NRF2 could be differentially regulated in these two types of tumours. NRF2 protein levels decreased significantly after chemotherapy in tumours characterized by major-H2AX decrease, while it remained unchanged in tumours with minor-H2AX decrease (Fig 5I). Consistently, the expression of several NRF2-target genes was significantly downregulated in tumours with major-H2AX decrease, compared to tumours with minor-H2AX decrease (Fig 5J). Taken as a whole, these data suggest that TNBC with impaired NRF2-dependent antioxidant response exhibits major-H2AX decrease. This could enhance chemosensitivity of tumour cells that enter into apoptosis. This is consistent with the better clinical outcome observed for TN patients with major H2AX

**Table 1.** **Main patient characteristics and clinico-pathological features of LumA, HER2 and TNBC patients, treated at the Institut Curie between 2000 and 2007.**

| Characteristics | Patients (*n* = 117) (%) |
| --- | --- |
| Subtype | |
| LumA | 33 (28) |
| Her2 | 37 (32) |
| TN | 47 (40) |
| Age at diagnosis (years) | |
| ≤ 50 | 41 (35) |
| > 50 | 64 (55) |
| NA | 12 (10) |
| Pathological tumour size pT | |
| pT1 | 48 (41) |
| pT2 | 45 (38) |
| pT3 | 8 (7) |
| pT4 | 4 (3) |
| NA | 12 (10) |
| Pathological lymph node status pN | |
| pN0 | 52 (44) |
| pN1 | 29 (25) |
| pN2 | 19 (16) |
| pN3 | 4 (3) |
| NA | 13 (11) |
| Metastasis status pM | |
| M0 | 103 (88) |
| M1 | 2 (2) |
| NA | 12 (10) |
| Histological Grade | |
| I–II | 29 (25) |
| III | 76 (65) |
| NA | 12 (10) |
| Tumour size (mm) | |
| < 20 | 38 (32) |
| [20–30] | 29 (25) |
| [30–50] | 19 (16) |
| ≥ 50 | 9 (8) |
| NA | 22 (19) |

Three classes of invasive ductal adenocarcinomas have been studied: Luminal-A (LumA), HER2-amplified adenocarcinomas (HER2) and Triple-Negative (TN) BC. A total of 33 LumA, 37 HER2 and 47 TNBC patients were included in the current study. HER2-amplified carcinomas have been defined according to the ERBB2 immunostaining using ASCO's guideline. LumA tumours were defined by positive immunostaining for ER (Estrogen receptor) and/or PR (Progesterone receptor). The cut-off used to define hormone receptor positivity was 10% of stained cells. Among invasive ductal carcinomas, the TN immunophenotype was defined as follows: ER⁻PR⁻ ERBB2⁻ with the expression of at least one of the following markers: KRT5/6⁺, EGF-R⁺, Kit⁺.

**Table 2.** **Clinical characteristics of TNBC patients treated with neo-adjuvant chemotherapy and features of the corresponding tumour samples.**

| Characteristics | Patients (*n* = 46) (%) |
| --- | --- |
| Subtype | |
| TN | 46 (100) |
| Age at diagnosis (years) | |
| ≤ 50 | 29 (63) |
| > 50 | 17 (37) |
| Clinical tumour size cT | |
| cT1 | 2 (4) |
| cT2 | 33 (72) |
| cT3 | 8 (17) |
| cT4 | 3 (7) |
| Clinical lymph node status cN | |
| cN0 | 19 (41) |
| cN1 | 19 (41) |
| cN2 | 2 (4) |
| NA | 6 (13) |
| Metastasis status M | |
| M0 | 46 (100) |
| M1 | 0 (0) |
| Histological Grade | |
| II | 9 (20) |
| III | 36 (78) |
| NA | 1 (2) |
| Tumour size (mm) | |
| < 20 | 2 (4) |
| [20–30] | 12 (26) |
| [30–50] | 19 (41) |
| ≥ 50 | 12 (26) |
| NA | 1 (2) |
| Therapeutic response | |
| Absent | 4 (9) |
| Minor | 13 (28) |
| Major | 7 (15) |
| Complete | 22 (48) |
| Chemotherapy | |
| 4EC, 4TXT | 41 (89) |
| 3EC, 4TXT | 5 (11) |

TN-invasive ductal carcinomas (ER⁻ PR⁻ HER2⁻) were diagnosed as described in the Methods section. Forty-six patients were selected for inclusion in this study. For all of them, the same treatment (see below) was used and the responses to anti-cancer therapy have been evaluated. Moreover, frozen and paraffin-embedded samples were available both before and after neo-adjuvant chemotherapy. All patients received epirubicin (75 mg/m²) and cyclophosphamide (750 mg/m²) intravenously every 3 weeks for four cycles followed by four additive cycles of docetaxel (100 mg/m²) every 3 weeks. Surgery was performed 21–45 days after cycle 8, according to the initial and post-chemotherapy assessment. The clinical response of each patient was evaluated by physicians according to the evolution of the tumour mass by monitoring patients throughout the chemotherapy and after surgery.

protein level decrease. In conclusion, our data indicate that successive cycles of chemotherapy in TN patients drive a chronic oxidative stress reaction associated with reduced levels of total H2AX protein, the extent of which could sensitize tumour cells to chemotherapy (Fig 6).

## Discussion

We uncover here a new ROS-dependent mechanism that reduces H2AX protein levels and increases chemosensitivity in aggressive BC. We detected a significant decrease in the total level of H2AX protein in pathophysiological conditions associated with loss of NRF2-dependent antioxidant activity and chronic oxidative stress, such as genetic inactivation of the antioxidant transcription factors JunD and NRF2, ageing and successive cycles of chemotherapy in TNBC patients. We also observed that the level of H2AX protein returned to normal after antioxidant use, demonstrating the central role of ROS in that regulation. We deciphered the molecular mechanism showing that persistent ROS target H2AX for degradation by the proteasome following its enhanced interaction with the E3 ubiquitin ligase RNF168. Importantly, ROS-mediated decrease in H2AX protein gives new insights on chemosensitivity of TNBC patients (see Fig 6 for model).

Although it is well established that TNBC patients exhibit a heterogeneous response to chemotherapy, the causes of this distinct chemosensitivity remain poorly understood. Here, we show that H2AX and γ-H2AX proteins follow a concomitant decrease in TNBC after chemotherapy. To our knowledge, this is the first time that H2AX downregulation is reported in TN breast carcinomas following treatment. Interestingly, major H2AX protein level decrease is associated with an impaired NRF2 transcriptomic response. Moreover, the extent of H2AX downregulation in these patients is a

reliable indicator of cancer cell apoptosis, tumour chemosensitivity and patient survival. Consistent with these findings, we show that partial depletion of H2AX in TNBC cells is sufficient to sensitize them to alkylating agents. Administration of chemotherapy induces significant DNA breaks that drive tumour cells with overwhelming damages towards apoptosis. In agreement with our observations, H2AX loss has been previously associated with increased genomic instability and enhanced sensitivity to additional acute stress. Indeed, H2AX haploinsufficiency due to the loss of a single *H2afx* allele compromises genomic integrity and enhances sensitivity to ionizing radiation (Bassing *et al*, 2002, 2003; Celeste *et al*, 2002, 2003). Moreover, H2AX knockout cells are highly sensitive to multiple DNA-damaging agents including cisplatin, camptothecin and radiation (Bassing *et al*, 2002; Meador *et al*, 2008; Revet *et al*, 2011). Similarly, miR-138 microRNA, known to inhibit H2AX, sensitizes cells to cisplatin and camptothecin, an effect that is reverted when H2AX is re-introduced into the cells (Wang *et al*, 2011). Finally, in recent phase II clinical trials, γ-H2AX was evaluated as a new pharmacodynamic marker of DNA damage produced by anti-cancer drugs (Löbrich *et al*, 2010; Redon *et al*, 2010; Wu *et al*, 2013). Based on its role in DNA repair, γ-H2AX is considered as a source of resistance to anti-neoplastic drugs and radiation therapy (Celeste *et al*, 2003; Meador *et al*, 2008; Nagelkerke *et al*, 2011; Revet *et al*, 2011; Matthaios *et al*, 2012). Indeed, the high capacity of forming DDR foci results in tumour resistance to chemotherapy (Asakawa *et al*, 2010; Guler *et al*, 2011; Nagelkerke *et al*, 2011) and γ-H2AX blocking peptides enhance cell death of irradiated resistant tumour cells (Taneja *et al*, 2004). Thus, total H2AX and γ-H2AX protein levels play a central role on cell viability when exposed to genotoxic stress. Tumour cells with DNA repair deficiencies, such as TNBC, have been associated with high levels of chromosomal rearrangements and are strikingly dependent on remaining DNA

---

**Figure 5.  Chemotherapy reduces H2AX protein levels.**

A  Gene set enrichment analysis (GSEA) performed by using NRF2-specific gene signature showing significant enrichment of this gene set in TN patients after chemotherapy. *P*-value refers to false discovery rate (FDR) q-value.

B  (left) Representative views of H2AX and γ-H2AX immunostaining in TN tumours (*N* = 22) before and after successive cycles of chemotherapy. Scale bars = 100 μm. Zoom views are focused on the epithelial cells. Scale bars = 25 μm. (right) Scatter plots of H2AX, γ-H2AX histological scores (Hscores) and γ-H2AX/H2AX ratio evaluated from immunostaining in the epithelium, before and after chemotherapy.

C  Correlation plot between Cisplatin IC50 (half maximal inhibitory concentration) and H2AX protein level defined by Western blots in BC cell lines (○MCF7, △BT474, ■MDA361, ▲SKBr3, ♦MDA453, ●MDA231, □HCC70). Correlation coefficients σ and *P*-value are based on Spearman's rank correlation test.

D  (up) Representative Western blot showing H2AX protein levels from whole cell extracts in HCC70 TNBC cell line transfected with non-targeting siRNA (siCtrl) or an siRNA directed against H2AX (siH2AX). Actin is used as internal control for protein loading. (down) Bar plot showing H2AX protein levels as assessed by densitometry analysis of Western blots (as shown above) (*n* = 3 independent experiments).

E  Bar plot showing the percentage (%) of apoptotic cells as assessed by flow cytometry analysis of Annexin V and DAPI staining in HCC70 cell line transfected with non-targeting siRNA (siCtrl) or an siRNA directed against H2AX (siH2AX) and next treated with cisplatin for 72 h.

F  Box-plot showing the impact of H2AX decrease on the therapeutic response of TNBC patients (*N* = 22). Reduction in H2AX protein level was assessed by the difference of H2AX Hscores before and after chemotherapy (as evaluated in B), for each patient (values in Log2). Only patients who have experienced an incomplete response have been taken into account. After treatment, H2AX staining was evaluated only on residual tumour cells. The therapeutic response was monitored by pathologists through evaluation of tumour residual mass and detection of tumour cells inside lymph nodes.

G  Kaplan–Meier curves showing specific overall survival (left) and progression-free survival (right) of TN patients with respect to major (*N* = 15) or minor (*N* = 7) H2AX protein level decrease, observed upon chemotherapy. Iterative analyses were performed on ΔLog2 H2AX Hscore to find optimal threshold that maximally discriminates the major and minor H2AX decrease (respectively, 2/3 and 1/3 of patients).

H  Scatter plots showing the number of cleaved caspase 3-positive cells/mm$^2$ in TNBC, categorized according to the major- or minor-H2AX protein level decrease after chemotherapy.

I  Scatter plots showing variation of NRF2 histological score in patients with major-H2AX decrease before and after chemotherapy.

J  Box-plots showing expression levels of representative NRF2-target genes in TN tumours with minor- or major-H2AX decrease, as indicated. Lines indicate Means, and whiskers show 10-90 percentiles. ACOT7: Acyl-CoA Thioesterase 7; PGD: Phosphogluconate Dehydrogenase. AHR: Aryl Hydrocarbon Receptor; JAG1: Jagged1. *P*-values are from *t*-test.

Data information: In all panels, data are shown as mean ± SEM (B, D, E, F and H). *P*-values are based on paired *t*-test (B and I), on Student's *t*-test (D, E, H and J) on Kruskal–Wallis test (F) and on Log-rank test (G). NS stands for not significant.

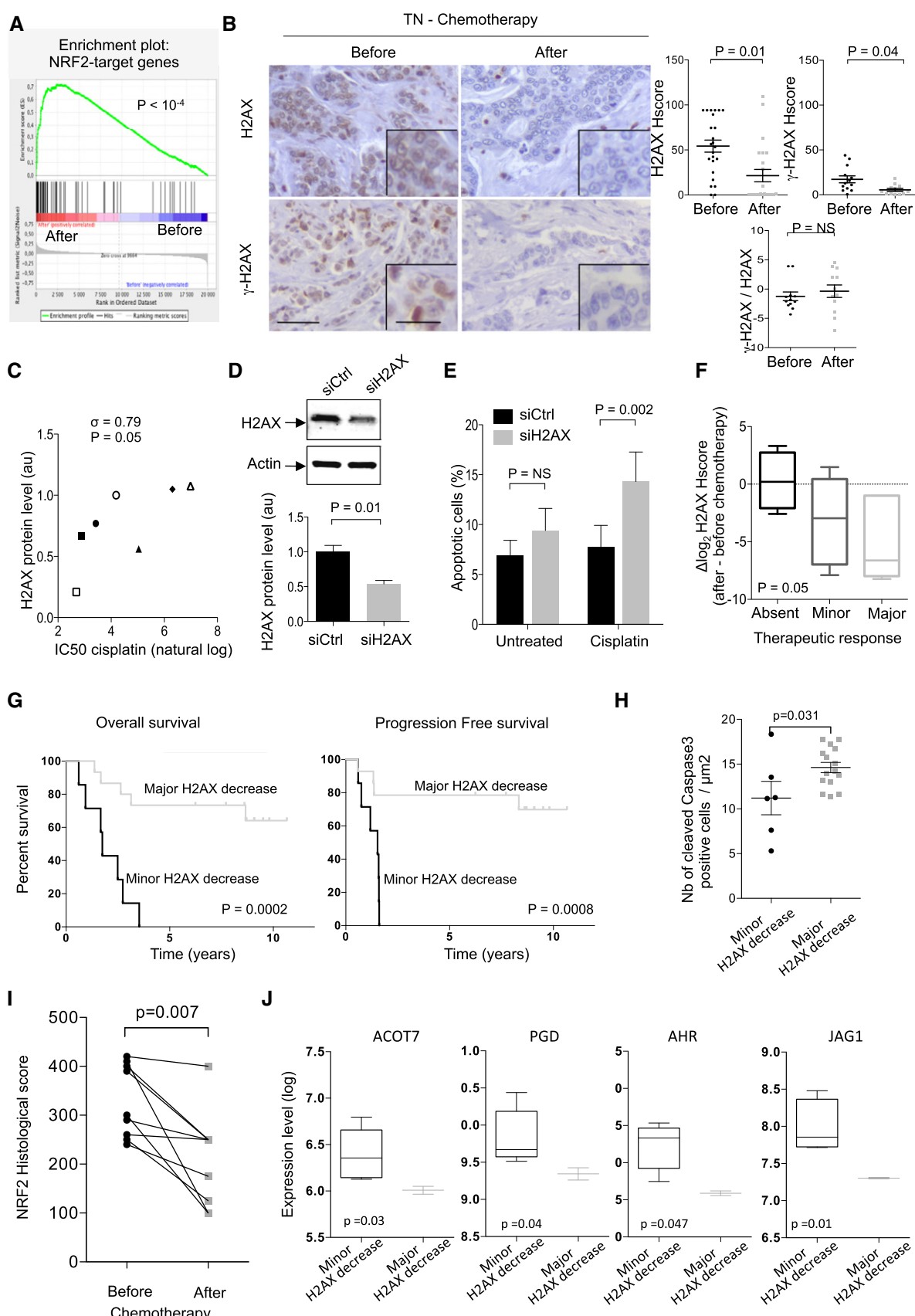

Figure 5.

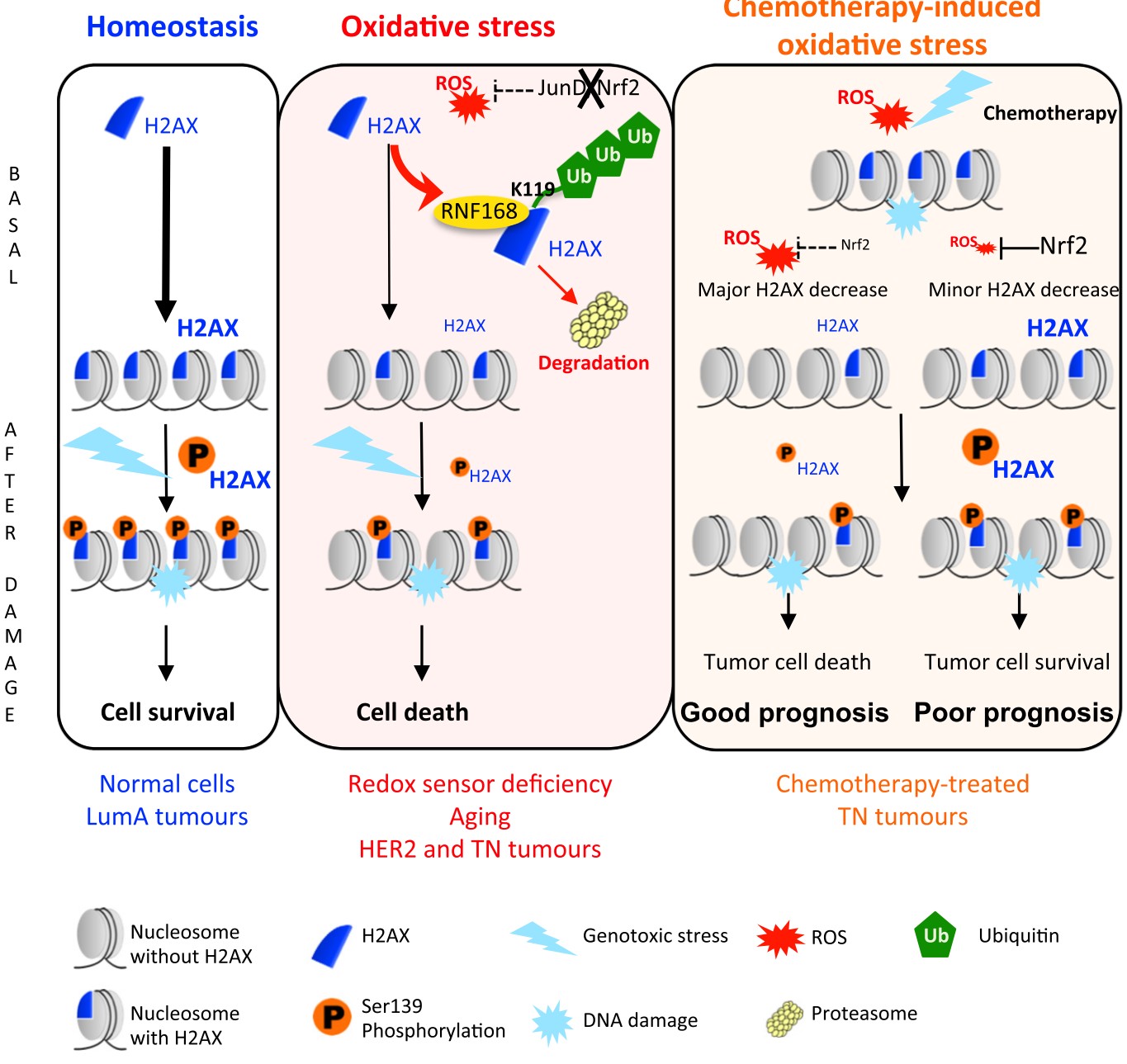

**Figure 6. Model.**

In normal cells (left panel), the newly synthesized histone variant H2AX is predominantly incorporated into the chromatin, where it acts as a major component of DNA repair pathway. Following DNA damage, H2AX is rapidly phosphorylated (hereafter referred to as γ-H2AX), γ-H2AX foci forms at the damaged sites and participate in DNA repair, which allows cell survival. In situations of chronic oxidative stress (middle panel), associated with reduced activity of the redox sensors JUND or NRF2 (due to junD or nfe2l2 genetic inactivation, ageing, aggressive BC or following chemotherapy), H2AX degradation is increased in the nucleoplasm resulting in reduced H2AX content at chromatin. Indeed, ROS induce an enhanced interaction with the E3 ubiquitin ligase RNF168, which indirectly leads to poly-ubiquitination of K119, followed by H2AX degradation by the proteasome. As a consequence, γ-H2AX levels are also reduced, cells become more sensitive to DNA-damaging agents and are prompted to die. Consistently, in TNBC (right panel), successive cycles of chemotherapy cause a persistent oxidative stress, which modulate NRF2 response and decrease H2AX and consequently γ-H2AX levels. This response varies from one patient to another: some patients show a defective NRF2 activity with an efficient downregulation of H2AX following chemotherapy (major H2AX decrease), while some others show a mild or absent reduction in H2AX protein levels (minor H2AX decrease). Tumours with major H2AX decrease are significantly enriched in oxidative stress signatures, demonstrating the role of redox imbalance in that process. In this group of patients, the efficient downregulation of the H2AX protein following chemotherapy prevents DNA repair and enhances tumour cell death. Consistently, these patients survive better than the ones with minor H2AX decrease, the latter exhibiting resistance to treatment.

repair activity for their survival (Natrajan *et al*, 2010; Bouwman & Jonkers, 2012). This genomic instability has been attributed mostly to mutations in *BRCA1* and *TP53* genes (Ellsworth *et al*, 2008; Kwei *et al*, 2010). The BRCA1/2 status has been determined in 20% of the TNBC patients analysed in the present study and no association was found between BRCA1 status and therapeutic response, arguing

that reduced H2AX levels might play an additional role independent of BRCA1. Finally, as mentioned above, TNBC patient with major H2AX decrease following treatment also shows a defective NRF2 anti-oxidant response. This indicates that TNBC cells exploit ROS to their advantage and are sensitive to reduction in antioxidant defences, in particular upon ROS-induced chemotherapy, a notion that could represent a certain clinical interest. Collectively, our data suggest that reduced antioxidant defences and subsequent decrease in total H2AX protein enhance sensitivity to DNA-damaging anti-cancer agents that is critical for TNBC patient survival.

In addition to its role in BC tumorigenesis and chemosensitivity, our work brings new insights into the role of RNF168, as it indicates that RNF168 not only participates in DDR signalling, as shown previously (Doil *et al*, 2009; Pinato *et al*, 2009; Stewart *et al*, 2009; Ismail *et al*, 2010; Ginjala *et al*, 2011; Gatti *et al*, 2012; Mattiroli *et al*, 2012; Bohgaki *et al*, 2013; Chen *et al*, 2013; Fradet-Turcotte *et al*, 2013; Leung *et al*, 2014; Kocylowski *et al*, 2015), but is also involved in H2AX stability under chronic stress. Indeed, the concerted action of the E3 ubiquitin ligases RNF8 and RNF168 upon DNA damage is well known to contribute to the activation of double strand break (DSB) repair pathway (Pinato *et al*, 2009; Mattiroli *et al*, 2012; Kocylowski *et al*, 2015). RNF8 activation at the site of DSB is required for the recruitment of RNF168, which specifically mono-ubiquitinates K13 and K15 residues of H2A-type histones and induces conjugation of K63-linked ubiquitin chains on these lysines to activate DDR signalling (Pinato *et al*, 2009; Mattiroli *et al*, 2012; Kocylowski *et al*, 2015). We establish that RNF168 and the K119 residue in H2AX are both important for H2AX degradation under chronic oxidative stress, while K13 or K15 single mutation and RNF8 silencing have no impact on H2AX protein stability. We thus assume that, in the context of chronic oxidative stress induced by the deletion of *junD* or *Nrf2*, RNF168 promotes H2AX ubiquitination on K119 in an indirect manner by a yet-unidentified ubiquitin ligase. As we analysed in our study K13 or K15 single mutants of H2AX, we could not exclude any compensation between these two lysine residues. Thus, under chronic oxidative stress, RNF168 might interact with K13–K15 residues of H2AX protein and further promote binding of a yet-unidentified ubiquitin ligase on K119, facilitating H2AX degradation. This model is in complete agreement with previously published results in the field (Pinato *et al*, 2009; Mattiroli *et al*, 2012; Kocylowski *et al*, 2015) and suggests for the first time a connection between RNF168 and K119 ubiquitination under chronic oxidative stress, which mechanism will be interesting to delineate in future work. Although RNF168-mediated K13–K15 ubiquitination is well established to facilitate 53BP1 binding to DSB sites, the role of K119 ubiquitination is less clear. Many studies have implicated this residue and the PRC1 complex in the DDR, but there still remains no strong mechanistic understanding of how this ubiquitylation functions within the DDR. Importantly, our work evaluates H2AX protein stability under chronic oxidative stress at steady state, and not after DNA damage. While most data on H2AX regulation have been established upon DDR signalling following acute genotoxic stress, the role of histone ubiquitination in chronic oxidative stress is poorly understood. The role of RNF168 in the regulation of H2AX stability under chronic oxidative stress is consistent with previously reported function of RNF168 mediating JMJD2A/KDM4A poly-ubiquitination and subsequent degradation by the proteasome (Mallette *et al*, 2012; Pinder *et al*, 2013). Interestingly, the JMJD2A/KDM4A protein degradation is required for proper signalling in early DDR, indicating that degradation of chromatin-linked proteins could be an essential mechanism for effective recruitment of key regulators to DNA-damaged sites. In a similar way, RNF168-dependent degradation of H2AX under chronic oxidative stress could be a reliable mechanism for the release of H2AX from nucleosome and turnover of newly synthetized H2AX protein at DNA-damaged sites. In that sense, while H2AX is ubiquitinated site-specifically by RNF168 and the PRC1 complex in the context of the nucleosome on chromatin following DSB signalling (Mattiroli *et al*, 2012, 2014; Leung *et al*, 2014), we observed that H2AX degradation is mostly detected in the nucleoplasmic fraction of *junD*-deficient cells. Consistent with this observation, H2AX in the nucleoplasm fraction has been shown to be less stable than when associated with chromatin (Liu *et al*, 2008; Rios-Doria *et al*, 2009). Moreover, RNF168 is able to ubiquitinate H2AX both into chromatin and nucleoplasm (Mattiroli *et al*, 2012). Taken as a whole, our work provides some clues about the mechanism by which chronic oxidative stress is involved in the regulation of H2AX protein turnover and brings new insights into the heterogeneous response of TNBC patients to chemotherapy.

## Materials and Methods

### Cell culture, chemical treatments, cell cycle and transfection

Due to premature senescence of *junD*-deficient MEFs, immortalized cell lines derived from *wt* or *junD*$^{-/-}$ embryos were generated using a conventional 3T3 protocol (Gerald *et al*, 2004).

*Nfe2l2*$^{-/-}$ cells were kindly provided by S. Biswal (Malhotra *et al*, 2010; Singh *et al*, 2013; Hayes & Dinkova-Kostova, 2014) and *Catalase*-deficient fibroblasts have kindly provided by Dr Marc Fransen, with authorization of Dr. Ye-Shih Ho, who generated them initially (Ho *et al*, 2004; Ivashchenko *et al*, 2011). All cell lines have been tested for the absence of mycoplasma contamination and were propagated in Dulbecco's modified Eagles medium (DMEM; Gibco) supplemented with 10% foetal bovine serum (FBS, PAA), penicillin (100 U/ml), streptomycin (100 μg/ml) (Gibco). For all data, we performed at least 3 independent experiments. For detecting endogenous H2AX proteins (Fig 3A–C), MG132 (Calbiochem #474791) and Cycloheximide (Sigma-Aldrich #C4859) treatments were applied for 8 h at a concentration of 10 μM and 75 μg/ml, respectively, on both *wt* and *junD*-deficient cells. Hydrogen peroxide (Sigma-Aldrich #H1009) and Camptothecin (SIGMA, #C9911) were applied at a concentration of 400 μM and 50 nM, respectively, for the indicated time. Cisplatin was applied at a concentration of 5 μM at 72 h post-transfection with siRNA. HCC70 BC cells were transfected with 10 nM of siRNA using 4 μl of DharmaFECT 1 (Dharmacon) per well in 2 ml final volume (6-well plates). Specific siRNA was used to knock down H2AX (5′-AACAACAAGAAGACGCGAATC-3′) and non-targeting siRNA was used as a negative control (Nontargeting AllStars Negative Control siRNA, Qiagen #1027280). Cisplatin IC50 tested on BC cell lines has been recently reported in (Garnett *et al*, 2012). Cell cycle distributions were performed on ethanol-fixed cells, stained with propidium iodide and analysed by flow cytometry. Flow cytometry data were acquired using CellQuest Pro

(Becton Dickinson) software on the FACSCalibur (Becton Dickinson) and were analysed using and ModfitLT (Verity) software. Cell cycle re-entry experiments were performed after 48 h of starvation in 0.25% FCS-containing medium. All treatments and transfection conditions used for expressing exogenous H2AX-FLAG are described below, # *Transfection and Immunoprecipitation (IP)*.

## Mouse strains and graft experiments

Due to previously reported male sterility (Thépot *et al*, 2000), *junD*-deficient mice were maintained through the breeding of heterozygous animals, as described in (Laurent *et al*, 2008). Genotyping was performed on DNA isolated from the tails of mice by PCR as described previously (Thépot *et al*, 2000). NAC (40 mM in drinking water, Sigma #A7250) was initiated during embryogenesis (via treatment of parent mice) and maintained throughout life; conditions preventing ageing phenotype of *junD*-deficient mice (Laurent *et al*, 2008) are used here. During all experiments, mice were maintained on standard diet containing 5% fat (Altromin #1314). Young versus old mice were defined as 2-month-old mice versus 18-month-old mice. Experiments in this paper have been performed on male mice. Animals were chosen in a randomized manner within each subgroup defined according to the genotype and age. For each condition, the sample size calculation was performed using "InvivoStat" software (http://invivostat.co.uk). The number of mice analysed in each experiment is indicated in the corresponding figure legend. Analyses were not performed under blinded conditions. All protocols involving mice and animal housing were in accordance with institutional guidelines as proposed by the French Ethics Committee and have been approved (Agreement number: CEEA-IC #118: 2013-06).

## Immunofluorescence

Fluorescence microscopy was performed as previously described with few modifications (Toullec *et al*, 2010). Briefly, cells were fixed in 4% paraformaldehyde for 20 min, permeabilized in 0.01% SDS for 10 min, rinsed in phosphate-buffered saline solutions (PBS) and blocked for 30 min in 10% FCS. Cells were stained with DAPI (50 μg/ml, Invitrogen #D1306) for DNA detection, together with specific antibody recognizing γ-H2AX (clone JBW301, 1/1500, Millipore #05-636) followed by fluorescein isothiocyanate (FITC)-coupled secondary antibody (1/300, Amersham #N1031). Slides were examined using a Zeiss Axioplan 2, and images were acquired with identical exposure times and settings using a digital camera (Photometrix Quantix). Fluorescence image analysis was performed using the ImageJ software (Rasband, WS, ImageJ, U.S National Institutes of Health, Bethesda, Maryland, USA, 1997–2008). Quantification of γ-H2AX large foci (diameter > 0.8 μm) was assessed by ImageJ software and further confirmed by visual inspection of images. Briefly, foci were automatically detected by selection of pixel with maximum intensity (Find Maxima macro) in the region of interest defined using DAPI staining. Detected foci had a diameter of at least 0.8 μm, which corresponds to large foci (Staaf *et al*, 2012). The number of foci per cell was evaluated using three independent cell lines for each condition. In average, 50 nuclei per genotype (with a minimum of 30 nuclei) have been used for quantification.

## Protein extracts and Western blot analysis

Whole cell extracts were performed using lysis buffer composed of 50 mM Tris pH 6.8, 2% SDS, 5% glycerol, 2 mM DTT, 2.5 mM EDTA, 2.5 mM EGTA, 2x Halt Phosphatase inhibitor (Perbio #78420), Protease inhibitor cocktail complete MINI EDTA-free (Roche #1836170), 4 mM $Na_3VO_4$ and 20 mM NaF. Heated buffer was directly applied on cells. After scratching, the solution is boiled at 95°C for 5 min. Sonication was applied for 10 min (cycles of 30 s ON—1 min 30 s OFF). The protein extract was then snap-frozen in liquid nitrogen and short-term stored at −80°C. Protein concentrations were evaluated using BCA assay (Thermo Scientific #23252). Free fraction extracts were performed in P300 lysis buffer composed of 20 mM $NaH_2PO_4$, 250 mM NaCl, 30 mM NaPPi, 5 mM EDTA, 0,1% NP-40, 5 mM DTT, 1× Halt Phosphatase inhibitor, Protease inhibitor cocktail complete MINI EDTA-free (Roche #1836170). Cells were lysed in P300 buffer for 30 min on ice. Two consecutive centrifugations were applied at a speed of 20,664 *g* at 4°C for 15 min each. Cells extracts were short-term stored at −80°C. Protein concentration was evaluated using Bradford assay (Bio-Rad #500-0006).

For immunoblotting analysis, H2AX being much easily visible on whole cell extracts enriched in chromatin-associated H2AX, the amount of total protein loaded in the gel is much higher for the free fraction extracts (15–20 μg) than for the whole cell extracts (3–5 μg). Western blotting was performed as previously described (Gerald *et al*, 2004). In brief, blots were incubated with horseradish peroxidase-conjugated secondary antibody (Amersham) followed by detection with enhanced chemoluminescence and exposed to autoradiography or with fluorochrome-conjugated secondary antibody followed by detection with the Odyssey Infrared Imaging System (LICOR Biosciences). The antibodies used are the following: H2AX (1/10,000, Abcam #ab11175), Jun (clone H-79, 1/1,000, Santa Cruz #sc-1694), γ-H2AX (clone JBW301, 1/1,000, Millipore #05-636), H2B (1/40,000, Abcam #ab1790), PKC delta phospho Y311 (1/500, Abcam #ab76181), Kap1 phospho S824 (1/1,000, Bethyl #A300-767A), actin (1/40,000, Sigma #A1978), RNF168 (1/1,000, Millipore # 06-1130), HA (1/10,000, Roche #11 867 423 001), FLAG (1/10,000, Sigma #F18-04).

## Transfection with H2AX-encoding vector and Immunoprecipitation (IP)

Constructs encoding mouse H2AX-WT and mutants have been kindly provided by Kyle Miller and were described previously in (Chen *et al*, 2013; Leung *et al*, 2014). In brief, a 3×Flag-tag was added to the N-terminus of the mouse *H2afx* gene by PCR and cloned into pENTR11 vector. 3×Flag-H2AX was Gateway cloned into pcDNA 6.2V5/DEST and is referred to as H2AX-WT vector in the current study. Indicated point mutations (K13A, K15A and K119A) were introduced into the H2AX-WT vector using the QuikChange mutagenesis kit (Stratagene) and correspond to the H2AX-K13, H2AX-K15 and H2AX-K119 constructs described here. 3×Flag-H2AX-9K to R was custom-synthesized and subsequently cloned in the same manner as 3×Flag-H2AX into the pcDNA 6.2V5/DEST vector. All constructs were fully sequenced before use for verifying mutations.

For co-transfection of H2AX-Flag and Ubiquitin-HA vectors, *wt* and *junD*$^{-/-}$ fibroblasts were seeded at $4 \times 10^5$ and $5 \times 10^5$ cells, respectively, into 10-cm Petri dishes and transfected with jetPEI transfection reagent according to the manufacturer's instructions with 1:2 ratio (# 101-10, Polyplus transfection™) in antibiotic-free DMEM, with 5 μg of H2AX-encoding plasmid (either H2AX-WT or H2AX mutant constructs) and 5 μg of pEBB-Ubiquitin-HA vector (a kind gift from Jacques Camonis). Cells were harvested 48 h after transfection, for immunoprecipitation (IP).

For IP, transfected cells were washed and harvested in cold PBS. The cells were spin down for 5 min at 367 *g* at 4°C and resuspended into p300 lysis buffer [20 mM NaH$_2$PO$_4$, 250 mM NaCl, 30 mM NaPPi (Na$_4$P$_2$O$_7$), 0.1% NP-40, 5 mM EDTA supplemented with proteases inhibitors (Roche #1836170) and Halt™ phosphatase inhibitor cocktail (Thermo Scientific #78426)]. Cells were incubated on ice for 30 min with three rounds of vortex and then sonicated at high power for 10 cycles (sonication cycle: 30 s ON and 30 s OFF) in a Bioptur® standard (Diagenode). The extracts were centrifuged at 16,000 *g* for 10 min at 4°C. The protein concentration of the supernatant was determined using the Bio-Rad D$_c$ Protein Assay Kit according to the manufacturer's instructions (Bio-Rad #500-0006). For immunoprecipitation, 250 μg of proteins was processed immediately and incubated on a wheel overnight at 4°C with 25 μl of anti-Flag M2 affinity gel (Sigma #A2220). The beads were washed four times with p300 lysis buffer. Lastly, 50 μl of Laemmli sample buffer (Bio-Rad #161-0737) supplemented with 100 mM DTT was added on top of the beads and boiled for 10 min at 95°C. Western blot analysis of IP samples was then performed as described above and probed with anti-HA (1/10,000, Roche #11 867 423 001), anti-Flag (1/10,000, Sigma #F18-04) or anti-RNF168 (1/1,000, Millipore # 06-1130) antibody.

### CHX treatment: determination of H2AX-Flag half-life

*wt* and *junD*$^{-/-}$ fibroblasts were seeded at $1 \times 10^5$ and $1.5 \times 10^5$ cells, respectively, into 6-well plates in 2 ml of antibiotic-free DMEM and transfected with 1 μg of H2AX-encoding plasmids (either H2AX-WT or H2AX-K119 constructs) with jetPRIME reagent (Polyplus transfection™ #114-15) according to the manufacturer's instructions. After 48 h of siRNA transfection, the cells were treated or not with Cycloheximide at 200 μg/ml (Sigma #CA4859) at various time points. Proteins were extracted directly in Laemmli sample buffer (Bio-Rad #161-0737) supplemented with 100 mM DTT, boiled for 10 min at 95°C and sonicated at high power for 10 cycles (sonication cycle: 30 s ON and 30 s OFF) in a Bioptur® standard (Diagenode). The extracts were centrifuged at 16,000 *g* for 10 min at 4°C. Western blot analysis of samples were then performed as described above and probed with anti-Flag antibody.

For double transfection, *wt* and *junD*$^{-/-}$ fibroblasts were seeded at $1 \times 10^5$ and $1.5 \times 10^5$ cells, respectively, into 6-well plates in 2 ml of antibiotic-free DMEM and transfected with 10 nM of siRNAs (either siControl #D-001810-02-20, siRNF168 #J-047329-11-0005, siRNF8 #L-048099-01-0005, siBMI1 #L-065526-01-0005, siUBE2N #L-064604-01-0005, Dharmacon) using DharmaFECT 1 transfection reagent according to the manufacturer's instructions (Dharmacon # T-2001-02). Forty-eight hours later, the cells again were transfected in the same medium with 1 μg of H2AX-encoding plasmids (either H2AX-WT or H2AX-K119 constructs) with

jetPRIME reagent (Polyplus transfection™ #114-15) according to the manufacturer's instructions. After 16 h of DNA transfection, the cells were treated or not with Cycloheximide at 200 μg/ml (Sigma #CA4859) at various time points. Proteins were extracted as described previously. Western blot analysis of samples were then performed as described above and probed with anti-Flag antibody.

For determining the half-life of H2AX-WT and H2AX-K119 proteins, densitometry analysis of Western blots were performed using the ImageJ software (Schneider *et al*, 2012). Quantifications were normalized to the untreated time point (*t0*) and H2AX protein half-life was evaluated from the degradation curve by extrapolating its linear part. Half-life of H2AX was calculated by the formula (0.5-b)/a, where "b" is the y-intercept and "a" the slope of the regression curve.

### Clonogenicity and cell survival assay

Clonogenicity experiments and data analysis were performed as described in (Franken *et al*, 2006). Right after γ-irradiation (2 or 5 Gy, as indicated), cells were seeded at low density (500–5,000 cells). After 7 days of incubation, cells were fixed and stained with 0.2% crystal violet in 20% ethanol during 20 min. After washing with water, the numbers of colonies containing at least 25 cells were counted. The number of surviving colonies was normalized to non-irradiated samples, with the latter value being set to 100%. This defines the surviving fraction (expressed in % with respect to the untreated samples).

### COMET assay

COMET assay was performed as described in (Quanz *et al*, 2009). Briefly, prior to treatment or after 10 Gy of γ-irradiation (*t* = 0 min, just after irradiation, or at *t* = 5, 15, 30 or 60 min following γ-irradiation), cells were suspended in 0.5% low-melting-point agarose and transferred onto a microscope slide pre-coated with agarose. Comets were performed in alkaline conditions. The parameters of the comets were quantified using the software Comet Assay 2 (Perceptive Instrument). Duplicate slides were processed for each experimental point. The tail moment was defined as the product of the percentage of DNA in the tail and the displacement between the head and the tail of the comet.

### Protein arrays

The reverse-phase protein array has been performed at the RPPA Platform of the Institut Curie. Experimental procedures have been performed, as in Troncale *et al* (2012). Briefly, serial diluted lysates were deposited onto nitrocellulose-covered slides, probed with primary antibodies and revealed with horseradish peroxidase-coupled secondary antibodies. Arrays were finally probed with Streptavidin-Alexa-647, dried and scanned using a GenePix 4000B microarray scanner (Molecular Devices). Spot intensity was determined with MicroVigene software (VigeneTech Inc). Data were normalized by Sypro Ruby protein stain. The antibodies used are listed here: H2AX (Abcam #ab11175), Phospho-53BP1 (Ser1778) (Cell Signaling Technology #2675), 53BP1 (Cell Signaling Technology #4937), Ape1 (Cell Signaling Technology #4128), ERCC1 (Cell Signaling Technology #3885), Cleaved PARP (Asp214) p25

(Epitomics #1051), PARP uncleaved p116 (Cell Signaling Technology #9532), Rad50 (Cell Signaling Technology #3427), Phospho-DNA-PK (#Ser2612) (Epitomics #23555-1), Phospho-FANCD2 (Ser222) (Cell Signaling Technology #4945), HSP90β (Cell Signaling Technology #5087S), Nbs1 (Cell Signaling Technology #3002).

The cell cycle control/DNA damage phospho-antibody microarray is a high-throughput ELISA based antibody array designed for semi-quantitative protein expression profiling, featuring 238 highly specific and well-characterized site-specific antibodies directed against 95 different proteins (Full Moon BioSystems # PCC238). Each antibody is printed with six replicates and covalently immobilized on polymer coated glass slides. Experimental procedures have been performed according to the manufacturer's instructions. Briefly, 75 μg of total biotinylated protein lysates extracted from *control*, *junD-* and *Nfe2l2*-deficient fibroblasts were conjugated to the antibody array. We then used Cy3-streptavidin dye for detection followed by GenePix 4000B microarray scanner (Molecular Devices) and MicroVigene software (VigeneTech Inc) for protein quantification.

### Apoptosis assays

At 72 h post-siRNA transfection, cells were treated with 5 μM of cisplatin for 24 h then harvested for apoptosis assay. Apoptosis was monitored by Annexin V (positive) and DAPI (negative) staining. Annexin V staining was performed using Annexin V-APC antibody (1/20, BD biosciences #561012) according to the manufacturer's instructions. DAPI was added at a final concentration of 1 μg/ml (Invitrogen #D1306).

### Description of cohorts of BC patients

The projects developed here are based on surgical residual tumour tissues available after histopathological analyses that are not needed for diagnostic purposes. There is no interference with the clinical practice. Analysis of tumour samples was performed according to the relevant national law on the protection of people taking part in biomedical research. All patients included in our study were informed by their referring oncologist that their biological samples could be used for research purposes and they gave their verbal informed consent. In case of patient refusal, which could be either orally expressed or written, residual tumour samples were not included in our study. The Institutional Review Board and Ethics committee of the Institut Curie Hospital Group approved all analyses realized in this study. Minimum sample size required to obtain a power of 80% and a significance level (Type I error probability) of 0.05 was computed using the R package pwr using the function pwr.t.test. HER2-amplified carcinomas have been defined according to ERBB2 immunostaining using ASCO's guideline. LumA tumours were defined by positive immunostaining for ER (Estrogen receptor) and/or PR (Progesterone receptor). The cut-off used to define hormone receptor positivity was 10% of stained cells. Among invasive ductal carcinomas, the TN immunophenotype was defined as follows: $ER^-PR^-ERBB2^-$ with the expression of at least one of the following markers: $KRT5/6^+$, $EGF-R^+$, $Kit^+$. Cohort characteristics have also been previously described in part in (Marty *et al*, 2008; Toullec *et al*, 2010; Maire *et al*, 2013) and are summarized in Table 1.

For studies based on the effect of neo-adjuvant chemotherapy, 46 patients were selected for inclusion in the study. For all of them, the same treatment (see below) was used and the responses to anti-cancer therapy have been evaluated. Moreover, frozen and paraffin-embedded samples were available, both before and after neo-adjuvant chemotherapy. All patients received epirubicin (75 mg/m$^2$) and cyclophosphamide (750 mg/m$^2$) intravenously every 3 weeks for four cycles followed by four additive cycles of docetaxel (100 mg/m$^2$) every 3 weeks. Surgery was performed 21–45 days after cycle 8, according to initial and post-chemotherapy assessment. The clinical therapeutic responses were evaluated by clinicians working at the Institut Curie according to the evolution of the tumour mass by monitoring patients through their chemotherapeutic treatment and after surgery. Specific overall survival, considering only patients who died from BC, and progression-free survival have been established with a clinical follow-up of 10 years. See also Table 2 for description of this cohort of TNBC patients.

### Immunohistochemistry on human breast carcinomas

Sections of paraffin-embedded tissue (3 μm) were stained using streptavidin-peroxidase protocol, immunostainer Benchmark, Ventana, Illkirch, France, with specific antibodies recognizing H2AX (1/200, Abcam #ab11175), γ-H2AX (1/100, Cell Signaling Technology #2577), actin, alpha2, smooth muscle (SMA) (1/400; Sigma #A2547), Cleaved Caspase 3 (1/150; Cell Signaling technology #9661) and NRF2 (1/100; Santa Cruz #sc-13032). Data were obtained using total sections from 10 LumA, 10 HER2 and 10 TN tumours, hybridized simultaneously. TMA (Tissue Micro-Array) from 33 LumA, 37 HER2 and 47 TN were used to confirm the data on a large set of patients. TMA was composed using three cores of tumours per case (1 mm of diameter each) and hybridized simultaneously. For studies based on the effect of neo-adjuvant chemotherapy, data were obtained using total sections from 46 TN patients. For staining quantification, at least five distinct areas of each tumour were evaluated by two different and independent investigators. A score, named Hscore for "Histological score", was given as a function of the percentage of positive cells and the staining intensity from 0 to 4. For H2AX and γ-H2AX staining, Hscores were established in both stromal fibroblasts and tumour epithelial cells. For all the other markers, Hscores were established in tumour epithelial cells. H2AX and γ-H2AX have been performed independently on two different cohorts of TNBC when compared to other BC subtypes (LumA, HER2, TNBC) or analysed regarding the impact of chemotherapy (TNBC patient samples pre- and post-chemotherapy). Therefore, we normalized the data between these two cohorts using a normalization factor, based on the H2AX mean Hscores in epithelial cells in these two TNBC cohorts. Mitotic index was defined as the number of mitotic figures found in 10 field areas, at a magnification ×400, and was evaluated by the pathologist staff working at the pathology department of the Institut Curie.

### Gene expression profiling in human BC

Only human tumours with a high content in epithelial tissue (at least 65%) have been used. Total RNA from LumA, HER2 and TN patients were extracted from frozen tumours with Trizol reagent (Life Technology, Inc.) and purified using the RNeasy MinElute

Cleanup kit (Qiagen #74204). RNA microarrays were conducted as described in (Marty *et al*, 2008). For studies based on TN patients before and after chemotherapy, tumours with an epithelial content of 50% in average were selected. TN subtypes in BC tissues before and after chemotherapy were purified using miRNeasy kit (Qiagen #217004). RNA quality was checked on an Agilent 2100 bioanalyser for both cohorts.

### High- versus low-H2AX BC

Accession number: The BC microarray data set is freely accessible in the Gene Expression Omnibus under the accession number: GSE45827.

Samples were hybridized on Affymetrix U133 plus 2.0 arrays. Data were first normalized using GC-RMA algorithm (R version 2.14.1; http://cran.r-project.org). After normalization, only genes with a log2 intensity > 4 in at least 5% of all samples were considered for the analysis. Samples were classified into "High" or "Low" subgroup according to the median value of the H2AX epithelial histological score (Hscore) within each tumour subtype (TN or HER2). Genes differentially expressed according to this classification were then identified using a Welch's *t*-test (*P*-value < 0.05 and fold change > 1.5). Functional analysis using DAVID software was finally performed in order to define the Gene Ontology pathways significantly enriched in "Low"- versus "High"-H2AX tumour samples. Fold enrichment is defined as the ratio between input genes and background information. *P*-values are calculated using a hypergeometric-based method (Huang da *et al*, 2009).

### Differentially expressed genes upon chemotherapy in TN patients

Accession number: The microarray data set from TN patients, before and after chemotherapy (excluding patients with complete response to treatment), is freely accessible in the Gene Expression Omnibus under the accession number: GSE43816.

TN ductal carcinomas were analysed on Human Gene 1.1ST arrays (Affymetrix), before and after chemotherapy. The data were analysed using Partek Genomic Suite version 6.6. Normalization was performed using RMA background correction with pre-background adjustment for GC content. Exons were summarized to genes using average method. Only probes with a log2 intensity > 5.2 in at least 5% of all samples were kept for further analysis. A differential analysis using a paired-samples *t*-test was applied in order to detect genes differentially expressed before and after neo-adjuvant chemotherapy within the same patient. DAVID functional annotation tool was then used to identify significant enriched Gene Ontology pathways significantly upregulated after chemotherapy. Fold enrichment is defined as the ratio between input genes and background information. *P*-values are calculated using a hypergeometric-based method (Huang da *et al*, 2009).

### Good- versus poor-therapeutic response in TNBC patients

Accession number: The microarray data set from TN patients before chemotherapy (including patients with complete response) is freely accessible in the Gene Expression Omnibus under the accession number: GSE45898. The biopsies (before treatment) were classified according to the clinical response of the patients: they have been classified into "Good" (major or complete) or "Poor" (absent or minor) therapeutic response subgroup. Genes differentially expressed between the two subgroups of patients were then

identified using a Welch's *t*-test (*P*-value < 0.05 and fold change > 1.5). Functional analysis using DAVID software was finally performed in order to detect the Gene Ontology pathways significantly enriched in patients with "Good" therapeutic response. "Fold enrichment" is defined as the ratio between input genes and background information. *P*-values are calculated using an hypergeometric-based method (Huang da *et al*, 2009).

### Accession numbers

GSE accession number for the gene expression profiling in Luminal-A, HER2 and Triple-Negative BC patient cohort is GSE45827. GSE accession number for the gene expression profiling in TNBC before and after chemotherapy is GSE43816, and for biopsies GSE45898. The microarray data sets will be freely available, using the accession numbers for above mentioned, at the publication time of the paper.

### Statistical analysis

All experiments were performed at least in triplicate. Data shown are means ± SEM (unless otherwise specified) from at least three independent experiments using adapted statistical test, as indicated in figure legends. Differences were considered to be statistically significant at values of $P \leq 0.05$. Edges of box-plot represent the 25th and the 75th percentiles; the solid line in the box presents the median value and the error bars indicate 90th and 10th percentiles from at least three independent experiments. The horizontal dark line on the scatter plots represents the mean and the error bars the SEM. Univariate Cox proportional hazards regression was conducted with SPSS 19.0 software using the enter method. Survival analyses were carried out using Kaplan–Meier method and log-Rank test in R. Log-Rank test using successive iterations was performed to find the optimal sample size thresholds that maximally discriminate the "major" and the "minor" H2AX decrease subgroups of patients. The optimal threshold in terms of overall survival corresponds to 2/3 of patients with major H2AX decrease and 1/3 of patient with minor H2AX decrease.

**Expanded View** for this article is available online.

### Acknowledgements

We thank G. Almouzni, J. Hall and P. Bertrand for fruitful discussion about the work and critical reviewing of the manuscript. We are grateful to B. Fourquet, C. Massabeau and B. Sigal-Zafrani for discussions on cohorts of BC patients following radiochemotherapy. We also thank B. Tesson for the transcriptomic data of cohort I, to E. Padoy, L. Fuhrmann, M. Nourieh and O. Chouchane-Mlik from the Pathology department of Institut Curie for their help and expertise in selecting immunohistological samples. We acknowledge X. Sastre-Garau, O. Mariani and the engineers of the CRB Institut Curie for her help in preparing the mRNA samples from patient tumours, A. Nicolas, R.Leclere and M. Richard-son from the experimental pathology platform of Institut Curie for providing tumour sections and D. Gentien from the Curie Affymetrix platform. *Nfe2l2*$^{-/-}$ cells were kindly provided by S. Biswal. *Catalase*-deficient fibroblasts have kindly provided by Dr Marc Fransen, with authorization of Dr. Ye-Shih Ho, who generated them initially. We thank G. Rieunier and M-H. Stern for phospho-Kap1 antibody. We thank L. de Koning, head of the RPPA platform. We are grateful to all members of the animal facilities of Institute Curie for their help-ful expertise. KMM was supported by the Cancer Prevention Research Institute

## The paper explained

### Problem

It is well established that H2AX is strongly activated upon acute stress and plays a key role in DNA damage. However, little is known about its turnover and how this turnover might affect DNA damage response and cell survival. Our work addresses that question in the context of chronic stress and gives new insights on the mechanism involved in regulating H2AX protein stability. Moreover, we analyse the physiological relevance of this regulation following chemotherapy, the successive cycles of which increase ROS in a chronic manner. This question is particularly relevant in Triple-Negative breast cancers (TNBC), which remains the most aggressive BC subtype with undoubted medical needs. TNBC patients exhibit a heterogeneous response to chemotherapy that is poorly understood or predictable. Our study addresses the question of how the variation in H2AX protein levels could modulate chemosensitivity in these patients.

### Results

Here, we uncover a new stress-dependent mechanism that affects H2AX protein stability and chemosensitivity in aggressive BC. We show a significant decrease in the total level of H2AX protein in various pathophysiological conditions associated with chronic oxidative stress, *i.e.* genetic inactivation of antioxidant enzymes, ageing, aggressive BC and successive cycles of chemotherapy in patients. We decipher the molecular mechanism showing that persistent ROS target H2AX for degradation by the proteasome following its enhanced interaction with the E3 ubiquitin ligase RNF168 and its subsequent polyubiquitination on the lysine residue K119. As a consequence of H2AX degradation, the DNA damage response decreases and cells are more sensitive to damaging agents. In TNBC patients, cycles of chemotherapy sustainably increase ROS levels and reduce H2AX protein levels. Interestingly, the extent of H2AX decrease after treatment is indicative of therapeutic efficiency and patient survival indicating that ROS-mediated H2AX decrease plays a crucial role in chemosensitivity of TNBC patients.

### Impact

Our work describes how chronic oxidative stress, due to NRF2-signalling deficiency, regulates H2AX protein turnover. We show that H2AX decrease following chemotherapy can sensitize TNBC cells to chemotherapy and we establish, for the first time, the relationship between H2AX and NRF2 regulation in tumours. The extent of H2AX decrease after chemotherapy is indicative of therapy efficiency and TNBC patient survival. Our study thus describes a new molecular mechanism by which oxidative stress sensitizes tumour cells to DNA damage in the settings of chemotherapy treatment and brings new insights into the heterogeneous response of TNBC patients to chemotherapy.

of Texas (CPRIT, R1116) and is a CPRIT scholar. This study was supported by grants from the Institut National de la Sante et de la Recherche Medicale (Inserm), the Institut Curie, the Ligue Nationale Contre le Cancer (LNCC) (Equipe labelisée), the Ministere de la Recherche et de l'Education Nationale, the French National Institut of Cancer (INCa) and the Fondation ARC.

## Author contributions

FM-G and TG participated in the conception and design of the experiments. TG, VM, MC, FD and BB performed the experiments. YK contributed to the statistical analyses of the data. AV-S provided human samples analysed in the study. The molecular classes of tumours were analysed by AV-S, based on immunohistochemistry results. Chemotherapy response available for cohort II was evaluated by AV-S. TD provided clinical data and TMA human samples. MD participated in the COMET analysis. KMM provided constructs encoding H2AX and brought his expertise in DNA damage response. FM-G supervised all the project and wrote the paper with participation of all authors.

## Conflict of interest

The authors declare that no conflict of interest exists.

## For more information

Gene ontology database:

http://www.geneontology.org/GO.downloads.ontology.shtml

F. Mechta-Grigoriou's laboratory:

http://u830.curie.fr/fr/genetique-et-biologie-des-cancers/equipes/
equipe-stress-et-cancer/equipe-stress-et-cancer-00101

Institut Curie Hospital:

http://curie.fr/en/soins/care/our-healthcare-offering

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
