## [Review Process File · EMBO Molecular Medicine]

Chronic oxidative stress promotes H2AX protein degradation and enhances chemosensitivity in breast cancer patients

Grusso Tina, Mieulet Virginie, Cardon Melissa, Bourachot Brigitte, Kieffer Yann, Devun Flavien, Dubois Thierry, Dutreix Marie, Vincent-Salomon Anne, Miller Kyle M. and Mechta-Grigoriou Fatima

Corresponding author: Fatima Mechta-Grigoriou, Institut Curie

Review timeline:	Submission date:	20 February 2014
	Editorial Decision:	24 February 2014
	Resubmission:	25 September 2015
	Editorial Decision:	19 October 2015
	Revision received:	07 January 2016
	Editorial Decision:	04 February 2016
	Appeal:	08 February 2016
	Editorial Decision:	08 February 2016
	Revision received:	15 February 2016
	Accepted:	18 February 2016

Transaction Report:

Editor: Roberto Buccione and Céline Carret

1st Editorial Decision

24 February 2014

Thank you for the submission of your manuscript "Chronic oxidative stress enhances chemosensitivity by reducing H2AX protein levels".

I have now had the opportunity to carefully read your paper and the related literature and I have also discussed it with my colleagues. I am afraid that we concluded that the manuscript is not well suited for publication in EMBO Molecular Medicine and have therefore decided not to proceed with peer review.

You find that in JunD^{-/-} mice compared to control, H2AX was lower in cells and organs (due to protein degradation in the nucleoplasmic fraction) and that N-acetylcysteine increased H2AX levels. You also find that in CAFs from HER2 and triple-negative (TN) breast cancer (BCa), H2AX protein was lower whereas NADPH-oxidases were up-regulated in low H2AX TN and HER2 BCa.

We appreciate that you find reduced H2AX levels and increased oxidative stress in residual tumours from TN patients who did not fully respond to chemotherapy and that the extent of H2AX down-regulation after chemotherapy correlated with response. This was matched by the finding in vitro, that BCa cells with low H2AX display high cisplatin sensitivity.

Although we acknowledge the potential interest of your findings, we find them rather more suited to a specialistic venue. We are in fact, not persuaded, also due to the descriptive nature of your findings and the lack of clear causal relationships and mechanistic analysis, that your manuscript provides the level of conceptual advance we would like to see in an EMBO Molecular Medicine article. We also appreciate the potential value of H2AX as a prognostic marker, but we feel that the validation required to draw such conclusion is lacking at this time.

I am sorry that I could not bring better news this time.

Resubmission

25 September 2015

I am pleased to submit our manuscript entitled "Chronic oxidative stress promotes H2AX protein degradation and enhances chemosensitivity in breast cancer patients", for consideration as a research article in *EMBO Molecular Medicine*. This was a great pleasure to meet you during your last visit in our Institute at Paris. As you suggested at that time, I would like to submit to your appreciation a completed version of our work, which deciphers a new mechanism of regulation of the DNA repair protein, H2AX. Very recently, I met Dr Roberto Buccione, who kindly proposed to receive the paper. But, as I did not have time to share with him our results in a so detailed manner as we did during your visit at Curie, and as you were enthusiastic, I thus send this work to you. Of course, I would be pleased to exchange with Dr. Buccione about this work, if one of you wish so. In our present work, we uncover a new mechanism of H2AX regulation by chronic oxidative stress, which modulates chemosensitivity of breast cancer patients.

H2AX is one of the most important DNA damage sensor. Its activation under acute oxidative stress has been extensively documented. However, little is known about its turnover and how this turnover might affect DNA damage response and cell survival. Our work shows that H2AX levels dramatically drop in cells subjected to chronic oxidative stress. Persistent accumulation of reactive oxygen species (ROS) targets H2AX protein for degradation by the proteasome, following its enhanced interaction with the E3 ubiquitin ligase RNF168 and its subsequent poly-ubiquitination. As a consequence γ -H2AX, the phosphorylated form of H2AX involved in DNA damage response, is reduced, further increasing genomic instability and chemosensitivity. Interestingly, we show that these findings are relevant for treatment efficiency in patients with triple-negative breast cancer (TN BC). Indeed, the extent of H2AX

decrease following successive cycles of chemotherapy, which increase ROS in a chronic manner, is indicative of TN BC patients survival. Hence, our work uncovers a new ROS-dependent mechanism regulating H2AX turnover, further providing insights into genetic instability and chemosensitivity in one of the most aggressive breast cancer subtype.

The data presented in the proposed manuscript reveal a new functional link between oxidative stress response, DNA damage pathways and sensitivity to chemotherapy. As we bring compelling evidence for this relationship and its patho-physiological consequences, we believe that this manuscript may be of interest to the readership of *EMBO Molecular Medicine*. In that matter, we can suggest Dr. Nancy Hynes, as editorial board member of reference. Moreover, you may consider contacting Dr. Paola Chiarugi, Dr. Alvaro Monteiro, Dr. Thanos Halazonetis and Dr. Penny Jeggo, as reviewers, for their competence in oxidative stress and DNA damage response, respectively. We hope you will agree to consider our work, and we look forward to hearing back from you.

2nd Editorial Decision

19 October 2015

Thank you for the submission of your manuscript to EMBO Molecular Medicine. We have now heard back from the two referees whom we asked to evaluate your manuscript. Although the referees find the study to be of potential interest, they also raise a number of concerns that must be addressed in the next final version of your article.

You will see from the comments pasted below that the referees find the study of potential interest while highlighting concerns of an overlapping nature. The main issue is that the models used have little relevance to TNBC and this particular point should be addressed and findings strengthen according to referee #1 suggestions point 9. This referee also recommends looking into other markers of DDR which would indeed improve the conclusiveness of the data. In addition, both referees provide constructive comments to generally make the study better and I would like to suggest to follow these recommendations as much as possible.

Given these, I would like to give you the opportunity to revise your manuscript, with the understanding that the referee concerns must be fully addressed and that acceptance of the manuscript would entail a second round of review. However, please note that that it is our journal's policy to allow only a single round of revision, and that acceptance or rejection of the manuscript will therefore depend on the completeness of your response and the satisfaction of the referees with it.

I look forward to seeing a revised form of your manuscript as soon as possible.

***** Reviewer's comments *****

Referee #1 (Comments on Novelty/Model System):

See point 9.

Referee #1 (Remarks):

This manuscript presents some interesting findings leading to the conclusion that chronic oxidative stress promotes H2AX protein degradation and enhances chemosensitivity in breast cancer patients. The results are potentially interesting, but their medical relevance is not completely clear. I am afraid that I don't think that this manuscript is appropriate for EMBO Molecular Medicine.

Specific Points:

1. The authors focused on H2AX after examining the protein levels of 8 DNA damage response proteins in wt, junD^{-/-} and Nfe2l2^{-/-} MEFs (Fig. S1). The panel of examined proteins is very small and the results shown in Fig. S1 are not that impressive in terms of robustness and magnitude of effect. For example, 53BP1 levels decrease in junD^{-/-} MEFs, but increase in Nfe2l2^{-/-} MEFs. Rad50 behaves like 53BP1. These results form the basis for focusing on H2AX in this study. However, this is not a solid basis. A much wider screen (at least 100 proteins and additional systems of chronic oxidative stress) is needed to make a convincing case that H2AX is a target worth focusing on.
2. Fig. 1A shows that H2AX levels are lower in immortalized junD^{-/-} fibroblasts compared to wt cells. The effect may be due to oxidative stress, based on the results with NAC (Fig. 1C), but other contributing mechanisms cannot be excluded, since junD affects the expression of many genes. More experiments with other systems inducing oxidative stress are needed.
3. The H2AX levels in the wt and junD^{-/-} fibroblasts seem variable. In the untreated (0 h) cells shown in Figs 1D, 1E and 1G, the levels of H2AX between the wt and junD^{-/-} fibroblasts are more than ~ 2-fold lower in the junD^{-/-} fibroblasts only in panel 1E.
4. In Fig. 2 the authors show a DNA repair defect in junD^{-/-} fibroblasts, but do not link this to reduced H2AX levels.
5. Fig. 3D shows that in whole cell extracts FLAG-tagged H2AX is more highly ubiquitinated in junD^{-/-} fibroblasts than in wt cells and that ubiquitination is dependent mostly on K119. The latter point makes sense, since ubiquitination of K119 is pervasive.
6. Figs 3F, 3G argue that RNF168 is responsible for K119 ubiquitination and for regulating H2AX levels in junD^{-/-} fibroblasts. These results are not clear. The depletion of RNF168 by siRNA is not strong enough. The effects in Fig. 3G are also not too strong. The co-IP in Fig. 3H lacks controls (cells not expressing H2AX-FLAG and IP with FLAG).
7. Fig. 4 examines a panel of breast cancers. H2AX levels are high in LumA epithelial cells, intermediate in HER2 cells and low in TN cells. This correlates inversely with mitotic index (Fig. S5).
8. Figs 5A, 5B show H2AX levels before and after chemotherapy in TN cancer. These are not easy experiments to perform, since one needs to study residual cells after chemotherapy. The H2AX scores in the epithelial cells before chemotherapy (Fig. 5B; average about 180) are higher than the ones shown in Fig. 4B (average about 70). This suggests significant variability in H2AX scores in TN cancers. After chemotherapy the average H2AX score is about 70 (Fig. 5B).
9. Fig. 5G shows that patient survival (TN cancers) relates to decreases in H2AX levels after chemotherapy. However, what is the evidence that the decrease in H2AX levels is responsible for patient survival versus the possibility that the changes in H2AX protein levels are secondary to the response to therapy? How about other markers? Mitotic index, apoptosis? How do these relate to patient survival? Also, were tumor stage or other clinically relevant parameters associated to the decrease in H2AX levels, thus explaining the link between H2AX levels and patient survival?

Referee #2 (Comments on Novelty/Model System):

This work indicates that H2AX a critical component of the DNA damage response is reduced by chronic oxidative stress and that this may be related to chemoresistance of triple negative breast cancers.

To study this effect the group decided to choose two models of chronic stress Nrf2-KO mice and JunD^{-/-} mice. Although these models do present enhanced oxidative stress the effects on a reduction of H2AX could be due to direct effects of Nrf2 and JunD deficiency and not oxidative stress. Simpler models such as catalase-KO would definitely be cleaner.

Another issues is that a clear connection between these models and TBNC was not made. Although both models show the same behavior in regards to H2AX and could be useful to determine the mechanism of H2AX reduction in the context of Nrf2^{-/-} and JunD^{-/-} deficiency there is no indication TBNC are Nrf2 or JunD deficient

Finally it is very interesting that at low dose H2O2 or radiation the induction of H2AX is intact. The problem is when high dose is used. This is very interesting in the face of current knowledge that "worse" breast tumors exploit oxidative stress to their advantage but are sensitive to oxidants or the reduction in antioxidant defenses. This should be discussed as this is one of the most interesting findings of this study

Referee #2 (Remarks):

This work indicates that H2AX a critical component of the DNA damage response is reduced by chronic oxidative stress and that this may be related to chemoresistance of triple negative breast cancers.

To study this effect the group decided to choose two models of chronic stress Nrf2-KO mice and JunD^{-/-} mice. Although these models do present enhanced oxidative stress the effects on a reduction of H2AX could be due to direct effects of Nrf2 and JunD deficiency and not oxidative stress. Simpler models such as catalase-KO would definitely be cleaner.

Another issues is that a clear connection between these models and TBNC was not made. Although both models show the same behavior in regards to H2AX and could be useful to determine the mechanism of H2AX reduction in the context of Nrf2^{-/-} and JunD^{-/-} deficiency there is no indication TBNC are Nrf2 or JunD deficient

Finally it is very interesting that at low dose H2O2 or radiation the induction of H2AX is intact. The problem is when high dose is used. This is very interesting in the face of current knowledge that "worse" breast tumors exploit oxidative stress to their advantage but are sensitive to oxidants or the reduction in antioxidant defenses. This should be discussed as this is one of the most interesting findings of this study

Detailed reply to the reviewers' comments

Submission of a revised manuscript (EMM-2015-05891): Gruosso et al., "Chronic oxidative stress promotes H2AX protein degradation and enhances chemosensitivity in breast cancer patients". For better visualization of the discussion, initial comments of the reviewers are indicated in blue and our answers are in Black.

Reviewer #1 (Remarks to the Author):

This manuscript presents some interesting findings leading to the conclusion that chronic oxidative stress promotes H2AX protein degradation and enhances chemosensitivity in breast cancer patients. The results are potentially interesting, but their medical relevance is not completely clear. I am afraid that I don't think that this manuscript is appropriate for EMBO Molecular Medicine.

We thank this reviewer for his/her constructive comments, along with a positive assessment of our work. We appreciated his/her suggestions, which have indeed contributed to the improvement of our manuscript. Please consider below the modifications we have done to specifically address referee's comments.

We have considered with great attention the first reviewer's comment concerning the medical relevance of our work. As recommended, we have highlighted in the new version of the text the medical input of our work, in particular with respect to the results obtained from the 2 cohorts of breast cancer patients, which have been described in more details in the **new Tables 1 & 2** and **new Figures 4 & 5**.

Thanks to close collaborations with clinicians, we had access to TN BC human samples collected before and after chemotherapy, with a clinical follow up of 10 years, which represent a quite rare collection of samples. Although it is well established that TN BC patients exhibit a heterogeneous response to chemotherapy, the causes of this distinct chemosensitivity remain poorly understood. Our findings provide new clues for the understanding of this heterogeneity. Indeed, we identify 2 classes of TNBC patients, which exhibit either a major decrease or a minor decrease of H2AX protein level after treatment. In agreement with the fact that reduced quantity of H2AX protein increases sensitivity of BC cells to chemotherapeutic agents (Figure 5C-E, confirming previous studies quoted in the new version of the manuscript), TN BC with major H2AX decrease show a higher number of apoptotic tumor cells, compared to TN BC with minor H2AX decrease (Figure 5H). Consistent with these observations, the extent of H2AX decrease following treatment is indicative of therapeutic efficiency and patient survival, but is not associated with other clinical parameters, such as mitotic index, histological grade, axillary or distant metastases (Figure 5G and corresponding description, p14). Interestingly, tumors exhibiting a major H2AX decrease after treatment have an impaired NRF2 anti-oxidant response. Taken as a whole, these data show that H2AX decrease following chemotherapy sensitize TN BC cells to chemotherapy and confirm the relationship between H2AX and NRF2 regulation in tumors. Our data thus identify new players in the chemosensitivity of TN BC patients and provide some clues in the understanding of therapeutic response heterogeneity. These new results have now been included in the **New Figure 5, p13-14**.

In addition to these new results and as recommended by the reviewer, we have also deeply modified the Discussion in the new version of our manuscript, to improve the clarity of our message in terms of medical relevance, **p15**.

1. The authors focused on H2AX after examining the protein levels of 8 DNA damage response proteins in wt, junD^{-/-} and Nfe2l2^{-/-} MEFs (Fig. S1). The panel of examined proteins is very small and the results shown in Fig. S1 are not that impressive in terms of robustness and magnitude of effect. For example, 53BP1 levels decrease in junD^{-/-} MEFs,

but increase in *Nfe2l2*^{-/-} MEFs. Rad50 behaves like 53BP1. These results form the basis for focusing on H2AX in this study. However, this is not a solid basis. A much wider screen (at least 100 proteins and additional systems of chronic oxidative stress) is needed to make a convincing case that H2AX is a target worth focusing on.

As requested, we have increased the number of DDR proteins analyzed in *JunD*- and *Nrf2*-deficient cells. As suggested by the reviewer, we analyzed more proteins by using an additive independent assay, corresponding to an antibody microarray, including a large number of DDR proteins. These new results confirm our previous data and give interesting complementary information, by showing additive results on other proteins, which had not been tested in the previous version. These new data are now shown in the new **Supplemental Table 1, p5 (last part of the paragraph)**. Moreover, considering reviewer's comment, we sought to introduce our interest for H2AX in a more direct and adapted manner **p5 (first part of the paragraph)**.

2. Fig. 1A shows that H2AX levels are lower in immortalized *junD*^{-/-} fibroblasts compared to wt cells. The effect may be due to oxidative stress, based on the results with NAC (Fig. 1C), but other contributing mechanisms cannot be excluded, since *junD* affects the expression of many genes. More experiments with other systems inducing oxidative stress are needed.

As suggested by the reviewer, we have tested another model of chronic oxidative stress. In that purpose, we have studied catalase-deficient fibroblasts, kindly provided by Dr Marc Fransen, with authorization from Dr. Ye-Shih Ho who generated them initially. In catalase-deficient cells, we have not observed a decrease in total H2AX protein levels, as seen in *JunD*- or *Nrf2*-deficient fibroblasts. These results suggest that H2AX decrease under chronic oxidative stress is more specifically associated with the loss of *JunD*/*Nrf2*-mediated anti-oxidant response. This is an interesting point that we uncovered thanks to this new experiment. We have thus included these data in the **new Supplemental Figure 1A** and described them **p5**.

Moreover, all along the text, we have now emphasized the idea that reduced H2AX is associated with a defective *JunD*/*Nrf2*-mediated anti-oxidant response. The link between H2AX decrease and chronic stress-associated with *JunD* or *Nrf2*-deficiency is further strengthened by the fact that both aging and chemotherapeutic treatment are associated with NRF-2-dependent anti-oxidant response. Indeed, we and others have previously demonstrated that aging is associated with ROS increase and progressive decline of Jun and NRF2 activity. These observations are now described in details in the **first part of the results, p5**. Interestingly, we also show, in the revised version of our manuscript, that chemotherapy is significantly associated with a NRF2-mediated transcriptomic response. Importantly, NRF2-target genes are differentially expressed in patients with major-H2AX decrease and characterized by a good clinical outcome, compared to patients with minor H2AX decrease and poor prognosis. These new results have now been included in the **new Figure 5A** and **Figure 5I,J** and described **p13-14**.

3. The H2AX levels in the wt and *junD*^{-/-} fibroblasts seem variable. In the untreated (0 h) cells shown in Figs 1D, 1E and 1G, the levels of H2AX between the wt and *junD*^{-/-} fibroblasts are more than ~ 2-fold lower in the *junD*^{-/-} fibroblasts only in panel 1E.

We thank the reviewer for this observation. We have now included another exposure of the same blot, in the new version of the text, in order to avoid the variation of signals from one part of the figure to the others. The main changes in blot exposure showing H2AX protein levels have been inserted in the **new Figure 1F** (previously 1E), as requested by the reviewer.

4. In Fig. 2 the authors show a DNA repair defect in *junD*^{-/-} fibroblasts, but do not link this to reduced H2AX levels.

As requested by the referee, we performed a rescue experiment in *junD*-deficient fibroblast. H2AX-overexpression in *junD*^{-/-} fibroblasts restores their survival rate, compared to control conditions, indicating that H2AX is a key component for the survival of these cells. This observation is **now described in Figure 2I** of the new version of the manuscript, **p8 (1st paragraph)**. These results are consistent with previous studies showing that reduced levels of H2AX protein (generated by H2AX haploinsufficiency, H2AX KO or expression of H2AX-targeting miRNA) were sufficient to significantly impair cell survival. These previous studies showing the impact of partial inactivation of *H2AFX* on cell survival are described in the new manuscript, to improve the clarity of our message, **p15, 2nd paragraph of the Discussion**.

5. Fig. 3D shows that in whole cell extracts FLAG-tagged H2AX is more highly ubiquitinated in *junD*^{-/-} fibroblasts than in wt cells and that ubiquitination is dependent mostly on K119. The latter point makes sense, since ubiquitination of K119 is pervasive.

We thank the reviewer for this positive assessment. We have now introduced this notion in the discussion part, **p18, middle page**.

6. Figs 3F, 3G argue that RNF168 is responsible for K119 ubiquitination and for regulating H2AX levels in *junD*^{-/-} fibroblasts. These results are not clear. The depletion of RNF168 by siRNA is not strong enough. The effects in Fig. 3G are also not too strong. The co-IP in Fig. 3H lacks controls (cells not expressing H2AX-FLAG and IP with FLAG).

As mentioned by the reviewer, the depletion of RNF168 is not total. While we performed several experiments in that purpose, unfortunately we did not improve the efficiency of RNF168 silencing. Although partial, this silencing is sufficient to observe an impact on H2AX ubiquitination, which has never been observed previously and we hope the reviewer will agree that the novelty of these results deserve interest. As requested, we have also included a control for the IP for Figure 3H, the controls have been included in the **new Supplementary Figure S2H**.

7. Fig. 4 examines a panel of breast cancers. H2AX levels are high in LumA epithelial cells, intermediate in HER2 cells and low in TN cells. This correlates inversely with mitotic index (Fig. S5).

As mentioned by the referee, there is a faint but significant inverse correlation between H2AX histological score and mitotic index, when considering all breast cancer subtypes together. In contrast, there is no association between H2AX protein level and mitotic index when considering each BC subtype separately. As recommended by the referee, these results have been added and described in more details in order to be as precise and accurate as possible. New results have been included **Figure 4G,H** and described **p12, middle page**.

8. Figs 5A, 5B show H2AX levels before and after chemotherapy in TN cancer. These are not easy experiments to perform, since one needs to study residual cells after chemotherapy. The H2AX scores in the epithelial cells before chemotherapy (Fig. 5B; average about 180) are higher than the ones shown in Fig. 4B (average about 70). This suggests significant variability in H2AX scores in TN cancers. After chemotherapy the average H2AX score is about 70 (Fig. 5B).

We thank the reviewer for taking into account the difficulty of this type of experiment, considering access to samples, rarity of cohort (TN BC patient samples pre- and post-chemotherapy) and required evaluation of staining in residual tumor cells. The results, shown in Figures 4 and 5, correspond to independent experiments performed on different human

tumor samples. Tumors and sections were different and the hybridizations were not performed at the same time. This is the reason why the intensity of the staining is distinct between these independent experiments. Considering reviewer's comment, we have applied a corrective factor for normalizing the experiments between them. This correction is now mentioned in the Material & Method' section, **p27 (#Immunohistochemistry on human breast carcinomas)**. It obviously does not change the results, while avoiding confusion and allowing easier comparison between experiments and from one figure to another. We thank the reviewer for his remark that significantly improves the manuscript.

9. Fig. 5G shows that patient survival (TN cancers) relates to decreases in H2AX levels after chemotherapy. However, what is the evidence that the decrease in H2AX levels is responsible for patient survival versus the possibility that the changes in H2AX protein levels are secondary to the response to therapy? How about other markers? Mitotic index, apoptosis? How do these relate to patient survival? Also, were tumor stage or other clinically relevant parameters associated to the decrease in H2AX levels, thus explaining the link between H2AX levels and patient survival?

As mentioned by the reviewer, in human tumors, our data are mainly based on significant correlations, mechanistic studies being difficult and limited on patient samples. Still, we provide data demonstrating the key role of H2AX protein decrease in BC cell genotoxic sensitivity in **Figure 5C-E, p13-14**. Moreover, as recommended, we performed additional analyses on these human tumors, and provided now complementary information. Indeed, we do agree that this set of TN BC represent a new and rare dataset deserving deep analysis. We have first compared transcriptomes from TN BC, before and after chemotherapy, and observed that NRF2-specific anti-oxidant signature is significantly enriched following chemotherapy (**Figure 5A, p13**). Based on this observation, we performed NRF2-specific immunohistochemistry and evaluated its protein levels before and after chemotherapy by histological scoring. Interestingly, we observed that NRF2 protein levels decrease significantly after chemotherapy, in tumors characterized by major-H2AX decrease, while it remains unchanged in tumors with minor-H2AX decrease. Consistently, NRF2-target genes are significantly downregulated in tumors with major-H2AX decrease. These data suggest that NRF2-dependent anti-oxidant response is reduced in tumors with major-H2AX decrease, and associated with good prognosis. Accordingly, while a major-H2AX decrease in tumors is neither linked to mitotic index nor to tumor stage, this group of tumors exhibits an increased apoptotic level, as evaluated by cleaved caspase3 immunostaining. These results are now described in details in the **new Figure 5F-J, p14**.

Referee #2 (Comments on Novelty/Model System):

This work indicates that H2AX a critical component of the DNA damage response is reduced by chronic oxidative stress and that this may be related to chemoresistance of triple negative breast cancers.

We are really grateful to Reviewer 2 for his/her first positive evaluation of our manuscript and interesting suggestions he/she made. We have addressed his/her comments and have listed below the modifications introduced in the text, hoping the modifications included and the additive data that we now provide will give him/her complete satisfaction.

To study this effect the group decided to choose two models of chronic stress NrF2-KO mice and JunD-/- mice. Although these models do present enhanced oxidative stress the effects on a reduction of H2AX could be due to direct effects of Nrf2 and JunD deficiency and not oxidative stress. Simpler models such as catalase-KO would definitely be cleaner.

As suggested by the reviewer, we have tested another model of chronic oxidative stress. In that purpose, we have studied catalase-deficient fibroblasts, kindly provided by Dr Marc

Fransen, with authorization from Dr. Ye-Shih Ho, who generated them initially. In catalase-deficient cells, we have not detected a decrease in total H2AX protein levels, as observed in *JunD*- or *Nrf2*-deficient fibroblasts. These results suggest that H2AX decrease under chronic oxidative stress is more specifically associated with the loss of JunD/Nrf2-mediated anti-oxidant response. This is an interesting point that we uncovered thanks to this new experiment. We have thus included these data in the **new Supplemental Figure 1A** and described them **p5**. Moreover, all along the text, we now have emphasized the idea that reduced H2AX is associated with a defective JunD/Nrf2-mediated anti-oxidant response. The link between H2AX decrease and chronic stress-associated with *JunD* or *Nrf2*-deficiency is further strengthened by the fact that both aging and chemotherapeutic treatment are associated with NRF-2-dependent anti-oxidant response. Indeed, we and others have previously demonstrated that aging is associated with ROS increase and progressive decline of Jun and NRF2 activity. These observations are now described in details in the **first part of the results, p5**. Interestingly, we also show, in the revised version of our manuscript, that chemotherapy is significantly associated with NRF2-mediated transcriptomic response. Importantly, NRF2-target genes are differentially expressed in patients with major-H2AX decrease and characterized by a good clinical outcome, compared to patients with minor H2AX decrease and poor prognosis. These new results have now been included in the **new Figure 5A and Figure 5I,J** and described **p13-14**.

Another issues is that a clear connection between these models and TBNC was not made. Although both models show the same behaviour in regards to H2AX and could be useful to determine the mechanism of H2AX reduction in the context of *Nrf2*^{-/-} and *JunD*^{-/-} deficiency there is no indication TBNC are *Nrf2* or *JunD* deficient

As recommended, we performed additional analyses, which provided complementary information on TN BC pre- and post-chemotherapy. We first compared transcriptomes from TN BC, before and after chemotherapy, and observed that NRF2-specific anti-oxidant signature is significantly enriched following chemotherapy (**Figure 5A, p13**). Based on this observation, we performed NRF2-specific immunohistochemistry and evaluated its protein level before and after chemotherapy by histological scoring. Interestingly, we observed that NRF2 protein levels decrease significantly after chemotherapy in tumors characterized by major-H2AX decrease, while it remains unchanged in tumors with minor-H2AX decrease. Consistently, NRF2-target genes are significantly downregulated in tumors with major-H2AX decrease. These data suggest that NRF2-dependent anti-oxidant response is reduced in tumors with major-H2AX decrease, and associated with good prognosis for patients. Accordingly, while a major-H2AX decrease in tumors is neither linked to mitotic index nor to tumor stage, this group of tumors exhibits an increased apoptotic level, as evaluated by cleaved caspase3 immunostaining. Taken as a whole, these data show that following chemotherapy H2AX decrease sensitize TN BC cells to chemotherapy and confirm the relationship between H2AX and NRF2 regulation in tumors. Our data thus identify new players in the chemosensitivity of TN BC patients and provide some clues for understanding therapeutic response heterogeneity. These new results are now described in details in the **new Figure 5F-J, p14**.

Finally it is very interesting that at low dose H2O2 or radiation the induction of H2AX is intact. The problem is when high dose is used. This is very interesting in the face of current knowledge that "worse" breast tumors exploit oxidative stress to their advantage but are sensitive to oxidants or the reduction in antioxidant defenses. This should be discussed as this is one of the most interesting findings of this study

We are grateful to the reviewer for this very interesting comment. We have now introduced this notion in the discussion part, **p16, end page**.

Thank you for the submission of your manuscript "Chronic oxidative stress promotes H2AX protein degradation and enhances chemosensitivity in breast cancer patients". We have now heard back from the referees who we asked to re-evaluate your article.

As you remember from the 1st round of reviews, referee 1 was already very critical, while the other referee less so. Unfortunately, referee 1 remains unsatisfied by the revised article and does not agree with the model put forth as unsupported by the (new) data. The clinical observation remains of interest, unfortunately, publication on its own would not be sufficient for EMBO Molecular Medicine.

I also would like to let you know that during the course of the decision making process we have involved an external additional advisor to help us and counsel on the best way forward.

As clear and conclusive insights into a novel clinically relevant observation is key for publication in EMBO Molecular Medicine, and together with the fact that we only allow one main round of revision, I am very sorry to say that we see no other choice than returning the article to you at this point with the message that we can not publish it in EMBO Molecular Medicine.

I am sorry that I cannot be more positive, but hope you will find a better home for your study soon.

***** Reviewer's comments *****

Referee #1 (Comments on Novelty/Model System):

junD/Nrf2 deficiency affects multiple processes, not only ROS

Referee #1 (Remarks):

In this manuscript, the authors propose the following pathway:

reduced junD/Nrf2 function -> reactive oxygen species (ROS) -> increased interaction of H2AX with RNF168 -> ubiquitylation of H2AX on K119 -> degradation of H2AX -> chemosensitivity

The new data provided by the authors did not strengthen the above model.

1. Catalase-deficient cells did not exhibit a reduction in H2AX levels, as predicted by the model (Fig. S1A). The authors argue that this "confirms that H2AX decrease is more tightly associated with defects in JunD/Nrf2-mediated anti-oxidant defence". I am not sure that "confirm" is the right verb to use. The model did not predict that ROS had to be due to reduced junD/Nrf2 function. In fact, the new data suggest that the decrease in H2AX levels is not due to ROS, but to some other downstream effector of junD/Nrf2 deficiency; a possibility that I had suggested in my original comments.

2. The mechanism by which ROS would enhance the interaction of RNF168 with H2AX is unclear. We know how RNF168 is recruited to chromatin in the vicinity of DNA DSBs (via RNF8-mediated ubiquitylation of histone H1). Does ROS lead to increased H1 ubiquitylation? And, if so, how? Via

RNF8? Also, we know that RNF168 ubiquitylates K15/K16 of H2AX, not K119, as proposed by the authors.

In conclusion, the mechanistic model proposed by the authors is not well-developed and contradicts other, carefully-controlled studies. Thus, I am not convinced that the model is correct.

The clinical observation that reduced H2AX levels in breast cancers is linked to chemosensitivity is more interesting and makes biological sense. Perhaps, these data on their own, could form a manuscript.

Referee #2 (Comments on Novelty/Model System):

All my concerns have been addressed

Referee #2 (Remarks):

In this revised version all my previous concerns were addressed.

Appeal

08 February 2016

Thank you for your message and your time. As we discussed on the phone, I would like to raise several issues concerning evaluation by Referee 1 that could be important for your final decision.

One first important point is that the reviewer asked for additional mechanism regarding Histone H1, which was not mentioned in the first round of review. Considering that only one round of review is allowed in EMBO Mol. Med. review process, we could not address this question. Moreover, although interesting, this question seems beyond the scope of our paper, because our study is strictly focused on the regulation of the histone variant H2AX.

I would like also to discuss the other point underlined by the reviewer about the generally accepted mechanism of action of the RNF168 Ubiquitin ligase. I would like to insist on the fact that our data are not contradictory to the generally admitted view in the field. In that sense, we have cited and discussed our data in the context of the commonly admitted model. We never consider in our text that RNF168 ubiquitinates H2AX on K119, as indicated by the Reviewer. In contrast, we mentioned in our Discussion that “RNF8 activation at the site of DSB is required for the recruitment of RNF168, which specifically mono-ubiquitinates K13 and K15 residues of H2A-type histones, such as H2AX, and induces conjugation of K63-linked ubiquitin chains on these lysine residues to activate DDR signalling”. Thus, we have placed our data in the commonly admitted view in the field, our results being compatible with this model. However, considering Reviewer 1’s comment, I

consider that our text could be confusing and I would be delighted to modify this part of the discussion to avoid any misunderstanding, if you wish.

Finally, although Referee 2 had overlapping concerns with Referee 1, he/she was fully supportive of our work. Moreover, both reviewers recognize the medical interest of our data. Thus, I would be grateful if you could reconsider your decision concerning our paper.

Please, do not hesitate to contact me directly by phone or by email, if you have additional questions.

Many thanks in advance for considering my request,

Looking forward to hearing from you soon

4th Editorial Decision

08 February 2016

Thank you for contacting me and raising the issue you spotted regarding referee 1 asking for additional mechanism on histone H1 that was however not requested earlier. As Referee 2 is indeed supportive of publication and had some overlapping concerns, we discussed within the team and agreed to change our decision.

I therefore am pleased to inform you that we will be able to accept your manuscript pending the following final amendments:

1) I would like you to particularly rephrase, and make obvious the point highlighted in your last letter regarding your results being supportive and not contradictory to the general accepted mechanism in the field. Please provide a point-by-point letter detailing your response to referee 1. incorporate them accordingly.

Please submit your revised manuscript within two weeks. I look forward to seeing a revised form of your manuscript as soon as possible.

2nd Revision - authors' response

15 February 2016

Detailed reply to the reviewers' comments

Revised manuscript (EMM-2015-05891-V3-Q): Grusso et al., "Chronic oxidative stress promotes H2AX protein degradation and enhances chemosensitivity in breast cancer patients".

We thank the editorial committee for considering our work suitable for publication in *EMBO Molecular Medicine*. As requested by the Editor, we have modified the text and re-phrased some sentences in the second part of the Discussion, for insisting on the fact that our results are well integrated in current literature and are not against the general accepted mechanism.

Reviewer #1 (Remarks):

1. Catalase-deficient cells did not exhibit a reduction in H2AX levels, as predicted by the model (Fig. S1A). The authors argue that this "confirms that H2AX decrease is more tightly associated with defects in JunD/Nrf2-mediated anti-oxidant defence". I am not sure that "confirm" is the right verb to use. The model did not predict that ROS had to be due to reduced JunD/Nrf2 function. In fact, the new data suggest that the decrease in H2AX levels is not due to ROS, but to some other downstream effector of JunD/Nrf2 deficiency; a possibility that I had suggested in my original comments.

As requested, we replaced "confirming" by "indicating" p5. We do agree that we can establish this statement thanks to this experiment. Thus, the new sentence is the following: "H2AX decrease was not observed in *catalase*-deficient cells, indicating that H2AX decrease is tightly associated with defects in JunD/Nrf2". We show that the decrease in H2AX protein is related to oxidative stress in Figure 1C. In that experiment, we applied to *JunD*-deficient mice an anti-oxidant treatment, N-acetyl-cysteine, well-known to reduce ROS content in animals (and also verified in our own lab, please see Laurent G. et al., *Cell Metabolism*, 2008; Toullec A., *EMBO Mol. Med.*, 2010). H2AX protein levels were restored in NAC-treated *JunD*^{-/-} mice to the same extent as in their *wt* counterparts. Moreover, a similar observation was made in WT mice upon aging. Thus, long-term anti-oxidant treatment prevented H2AX down-regulation in *JunD*^{-/-} mice and in WT mice upon aging, supporting that ROS are involved in regulating the total level of H2AX protein *in vivo*.

2. The mechanism by which ROS would enhance the interaction of RNF168 with H2AX is unclear. We know how RNF168 is recruited to chromatin in the vicinity of DNA DSBs (via RNF8-mediated ubiquitylation of histone H1). Does ROS lead to increased H1 ubiquitylation? And, if so, how? Via RNF8? Also, we know that RNF168 ubiquitylates K15/K16 of H2AX, not K119, as proposed by the authors.

In conclusion, the mechanistic model proposed by the authors is not well-developed and contradicts other, carefully-controlled studies. Thus, I am not convinced that the model is correct.

In the initial review, Referee 1 was concerned by the medical interest of our work and he/she didn't ask any question regarding the mechanism of H2AX degradation that we described in Figure 3. The main issue raised by the Reviewer on this Figure was the lack of a control in the immunoprecipitation experiment. We agreed and provided this control in the revised version of the manuscript. Moreover, as requested, we improved siRNA efficiency and showed, as mentioned in the revised version of the manuscript, that siRNF168 silencing, although partial, is sufficient to significantly stabilize H2AX protein in *JunD*-deficient cells. Referee 1 didn't ask any question regarding Histone H1 in the first review. This question is interesting, particularly in the context of the recent publication from Niels Mailand (Thorslund T et al., *Nature*, 2015). This work shows that Histone H1 is targeted at DSB by RNF8, which promotes binding of RNF168 to its ubiquitin targets. Indeed, RNF8 mediated UBC13/UBE2N-

dependent K63-linked ubiquitination of Histone H1 at DSB. Thus, this study identifies histone H1 as a key target of RNF8-UBC13 in DSB signalling. Although interesting, the reviewer in the first round of review has not mentioned questions regarding histone H1. Moreover, as our work is strictly focused on H2AX and not on the broader role of DDR signalling in this pathway, we thus consider that this question is beyond the scope of this paper and should be addressed in follow up studies.

In the second part of the discussion (p17-18), we comment in depth about our results in the context of current literature. We never consider in our text that RNF168 ubiquitinates H2AX on K119, as indicated by the Reviewer. In contrast, we mention (p17) that “RNF8 activation at the site of DSB is required for the recruitment of RNF168, which specifically mono-ubiquitinates K13 and K15 residues of H2A-type histones, such as H2AX, and induces conjugation of K63-linked ubiquitin chains on these lysine residues to activate DDR signalling”. Thus, we cite and discuss our data in the context of the mechanistic model commonly admitted, i.e. RNF168 catalyses H2AX ubiquitination on K13 and K15 at DSB to activate DNA repair signalling pathways. We establish that RNF168 and the K119 residue in H2AX are both important for H2AX degradation under chronic oxidative stress, while single K13 or K15 mutation has no impact on H2AX stability. We thus assume that, in the context of chronic oxidative stress induced by the deletion of *junD* or *Nrf2*, RNF168 promotes H2AX ubiquitination on K119, in an indirect manner, by a yet-unidentified ubiquitin-ligase. As we analysed in our study K13 or K15 H2AX single mutants, we could not exclude any compensation between these two Lysine residues. Thus, under chronic oxidative stress, RNF168 might interact with K13-K15 residues in the H2AX protein and further promote binding of a yet-unidentified ubiquitin-ligase on K119, facilitating H2AX degradation. Considering Reviewer 1’s comment, we have modified the second part of the discussion to avoid any misunderstanding on the integration of our data in the current literature.

At last, we emphasize in the discussion that we study the regulation of H2AX protein stability under chronic oxidative stress at steady state, and not after DNA damage. In contrast, most data on H2AX regulation have been established upon DDR signalling following an acute genotoxic stress, which could differentially impact H2AX regulation. This is an important distinction that we have highlighted in the manuscript to avoid any confusion. For instance, it is noteworthy that H2AX degradation under oxidative stress occurs mainly in the nucleoplasm, while H2AX is ubiquitinated site-specifically by RNF168 and the PRC1 complex in the context of the nucleosome on chromatin following DSB signalling. Thus, we feel it is important to place our results in this perspective in the discussion.

The clinical observation that reduced H2AX levels in breast cancers is linked to chemosensitivity is more interesting and makes biological sense. Perhaps, these data on their own, could form a manuscript.

We thank the reviewer for considering these data interesting and deserving publication.

Referee #2 (Remarks):

In this revised version all my previous concerns were addressed.

We thank the reviewer. We have greatly appreciated his/her previous concerns, as Referee 2’s comments have contributed to improve the quality of our manuscript and enhance our message.

Corresponding Author Name: MECHTA-GRIGORIOU Fatima

Manuscript Number: EMM-2015-05891-V2